# Nested leave-two-out cross-validation for the optimal crop yield model selection

Thi Lan Anh Dinh[1] and Filipe Aires[1]

[1]Sorbonne Université, Observatoire de Paris, Université PSL, CNRS, LERMA, 75014 Paris, France

**Correspondence:** Thi Lan Anh Dinh (lan-anh.dinh@obspm.fr)

**Abstract.** The use of statistical models to study the impact of weather on crop yield has not ceased to increase. Unfortunately, this type of application is characterised by datasets with a very limited number of samples (typically one sample per year). In general, statistical inference uses three datasets: the training dataset to optimise the model parameters, the validation dataset to select the best model, and the testing dataset to evaluate the model generalisation ability. Splitting the overall database into three datasets is often impossible in crop yield modelling due to the limited number of samples. The leave-one-out cross-validation method, or simply Leave-One-Out (LOO), is often used to assess model performance or to select among competing models when the sample size is small. However, the model choice is typically made using only the testing dataset, which can be misleading by favouring unnecessarily complex models. The nested cross-validation approach was introduced in machine learning to avoid this problem by truly utilising three datasets even with limited databases. In this study, we propose one particular implementation of the nested cross-validation, called the nested leave-two-out cross-validation method or simply the Leave-Two-Out (LTO), to choose the best model with an optimal model selection (using the validation dataset) and estimate the true model quality (using the testing dataset). Two applications are considered: Robusta coffee in Cu M'gar (Dak Lak, Vietnam) and grain maize over 96 French departments. In both cases, LOO is misleading by choosing too complex models; LTO indicates that simpler models actually perform better when a reliable generalisation test is considered. The simple models obtained using the LTO approach have improved yield anomaly forecasting skills in both study crops. This LTO approach can also be used in seasonal forecasting applications. We suggest that the LTO method should become a standard procedure for statistical crop modelling.

## 1 Introduction

Many approaches are available to study the impact of climate and weather variables on crop yield. Statistical modelling, which aims to find relations between a set of explanatory variables and crop yield, is a widely used approach (see, for example, Lobell and Burke (2010); Mathieu and Aires (2016); Gornott and Wechsung (2016); Kern et al. (2018)). This approach has many advantages, such as identifying crop production sensitivities (Mathieu and Aires, 2018a), complementing field experiments (Gaudio et al., 2019), and helping in adaptation strategies (Iizumi et al., 2013), but it is often complex to understand and to use for several reasons.

Unfortunately, crop modelling is often characterised by datasets with a very limited number of samples. For instance, Prasad et al. (2006) built a crop yield estimation model with 19 years of yield data. Ceglar et al. (2016) studied the impact of meteorological drivers over 26 years on grain maize and winter wheat yield in France. One year of data represents one sample in these applications, and about 20 samples are small for a data-driven approach. Small sample size poses two challenges to crop modelers. First, it makes it hard to choose among competing models. Second, it makes it hard to assess the quality of the chosen

model. For example, increasing the model complexity usually increases the goodness-of-fit of the model. However, it can lead to "overfitting" if the model is too complex and if we have a limited information included in the database. Overfitting occurs when the model fits the training dataset artificially well, but it cannot make good predictions on unseen data. To overcome these issues, in statistical modelling, the overall database is divided into three datasets: the training dataset to optimise the model parameters, the validation dataset to select the best model, and the testing dataset to evaluate the model generalisation ability

(Ripley, 1996). Splitting a small number of samples into three datasets is not easy.

Cross-validation (Allen, 1974; Stone, 1974) was introduced as an effective method for both model assessment and model selection when the data is relatively small. A common type of cross-validation is the Leave-One-Out cross-validation (LOO) that has been used in many crop models (Kogan et al., 2013; Zhao et al., 2018; Li et al., 2019). This approach relies on two datasets: a training dataset is used to calibrate the model, and a testing dataset is used to assess its quality. However, an

important drawback of LOO is that it uses the testing dataset to select the best model, which we assert is a bad practice, as we shall explain. Since the chosen model is not independent of the testing dataset, the obtained testing score may be unreliable. This is not a problem if there are many available samples. However, a small sample size can cause many issues: the model can overfit the training dataset; thus, the chosen model is not adequate, and our assessment of its generalisation ability is false. This mistake is often seen in crop modelling when overly complex models are developed based on a limited number

of samples (Jayakumar et al., 2016; de Oliveira Aparecido et al., 2017; Niedbała, 2018). Some regularisation techniques (e.g. information content techniques or dimension reduction techniques) can help to constrain models toward lower complexity to limit the overfitting problem (Lecerf et al., 2019). However, these approaches can become more technical and more challenging to interpret, especially for non-statisticians.

To solve the issues of LOO, another more complex approach has been introduced: nested cross-validation (Stone, 1974),

also known as double cross-validation or $k \times l$-fold cross-validation, is able to use three datasets: training, validation, and testing. In detail, this approach considers one inner loop cross-validation nested in an outer cross-validation loop. The inner loop is to select the best model (validation dataset), while the outer loop is to estimate its generalisation score (testing dataset). We found very few applications of this approach in the literature on statistical crop modelling (Laudien et al., 2020; Meroni et al., 2021; Laudien et al., 2022). This study proposes one particular implementation of this nested cross-validation (or $k \times l$-

55    fold cross-validation when $l = k-1$) called the Leave-Two-Out (LTO). Here, we used the LTO for three purposes: first, to obtain a reliable assessment of the model generalisation ability; second, to compare the performances of different predictive models; and third, to determine the optimal complexity of the statistical crop models. This approach is tested in two real-world applications: Robusta coffee in Cu M'gar (a district of Dak Lak province in Vietnam) from 2000 to 2018 and grain maize over 96 departments (i.e. administrative units) in France for the 1989-2010 period. The following sections of this study will

(1) introduce the materials and databases used for statistical crop models, (2) describe the role of three datasets in statistical inference, (3) introduce the two cross-validation approaches, (4) evaluate and select the "best model" by using LOO and LTO approaches, (5) estimate the Robusta coffee yield anomalies in Cu M'gar (Dak Lak, Vietnam), and (6) assess the seasonal yield anomaly forecasts for grain maize in France.

## 2  Modeling crop yield using machine learning

### 65  2.1  Materials

#### 2.1.1  Robusta coffee

**Overview**

Robusta ($Coffea\ canephora$) is among the two most widely-cultivated coffee species (the other being $Coffea\ arabica$, known as Arabica). About 40 % of the world's Robusta coffee is produced in the Central Highlands of Vietnam (USDA, 2019; 70  FAO, 2019) due to its adequate conditions in terms of elevation (200-1500 m), soil type (basalt soil), and climate (an annual average temperature of about 22 °C). In addition, intensive agricultural practices are used (e.g. fertilisation, irrigation, shade management, and pruning) in these coffee farms (Amarasinghe et al., 2015; Kath et al., 2020). The Central Highlands region includes four main coffee-producing provinces, and each province is divided into several districts. Here, we focus on Robusta coffee in Cu M'gar, one major coffee-producing district in the Central Highlands.

A coffee tree is a perennial, which is highly productive for about 30 years (Wintgens, 2004) but can be much longer (more than 50 years) with good management practices. Mature coffee trees undergo several stages before harvesting, including the vegetative stage (bud development) and the productive stage (flowering, fruit development, and maturation) (Wintgens, 2004). It requires about eight months (May to December) for the vegetative stage and about 9-11 months (January to September/November) from flowering until fruit ripening for Robusta coffee. The weather during the last few months before harvest 80  (i.e. the productive stage) is decisive for the yield (Craparo et al., 2015b; Kath et al., 2020), however, it has been shown that the weather during the previous year's growing season (i.e. the vegetative stage) has a big impact. A prolonged rainy season (14-19 months before harvest) favours vegetative growth and thus increases the potential coffee yield (Kath et al., 2021). As a result, it is necessary to consider the weather variables during both vegetative and productive stages when studying the weather impact on coffee yield. Thus, for this study, we analysed the weather of 19 months (from May of the previous year to November) 85  preceding the harvest.

**Yield database**

The Robusta coffee yield data were obtained from the General Statistics Office of Vietnam for the 2000-2018 period ($n_{samp} = 19$). We focus on Cu M'gar district as it is a leading coffee-producing district in Vietnam, accounting for about 10 % of Vietnam's total coffee production (i.e. 76400 tons for the 2000-2018 average).

The long-term trend represents the slow evolution of the crop yield; it often describes the changes in management like fertilisation or irrigation. Thus, suppressing this trend from the yield time series allows removing the (Mathieu and Aires,

2016). For Robusta coffee, a simple linear function is used to define the yield trend: $\overline{y}(t) = y_0 + \alpha t$, where $\overline{y}(t)$ is the long-term trend, $y_0$ is the yield in 2000, and $\alpha$ is the constant annual rate of change. Once the yield trend is defined, the coffee yield anomalies are calculated by removing this trend from the raw yield data. The Robusta coffee yield for year $t$ is noted as $y(t)$ and the coffee yield anomaly $a(t)$ (in %) is calculated as:

$$a(t) = \frac{y(t) - \overline{y}(t)}{\overline{y}(t)} \times 100. \tag{1}$$

If $a(t) > 0$, then the yield in year $t$ is higher than in a regular year, and vice versa. For example, an anomaly of $a(t) = -16$ implies that the yield for year $t$ is 16 % lower than the annual trend.

### 2.1.2 Grain maize

**Overview**

Grain maize ($Zea\ mays$) is among the most common annual crops in Europe. France, our study region, is the largest grain maize producer in Europe (EUROSTAT, 2021). The study area has been improved a lot in agro-management and irrigation practices after 1960, e.g. irrigation acres was about 50 % at the beginning of the $21^{st}$ century (Siebert et al., 2015; Schauberger et al., 2018; Ceglar et al., 2020). Although the sowing time varies for different regions (Olesen et al., 2012), the average growing season of French grain maize ranges from April to September (Ceglar et al., 2017; Agri4cast, 2021). Many previous studies showed that grain maize yield is sensitive to weather conditions (Ceglar et al., 2016, 2017; Lecerf et al., 2019), especially during crop growing season. Therefore, we will analyse the relationship of maize yield to meteorologic variables during the 6-month growing season.

**Yield database**

The French grain maize data (area, production, and yield) on the regional level (i.e. department which is an administrative unit in France) were collected from the Agreste website (https://agreste.agriculture.gouv.fr; "Statistique agricole annuelle ") for a period of 22 years (from 1989 to 2010). Here, we have modelled the yield of grain maize in 96 French departments (Fig. 1). Some specific tests (in Sect. 5) will focus on ten departments (as presented in Fig. 1(d)) where the average grain maize production is higher than $4 \times 10^5$ tons (or the area is higher than 40 thousand hectares).

Similar to Robusta coffee, the grain maize anomalies are calculated by removing the long-term yield trend. Here, a 10-year moving average window is used because the trend is slightly more complex than the one we found for Robusta coffee.

### 2.1.3 Weather database

The monthly-mean total precipitation (P) and 2 m temperature (T) variables were collected for the period 1981-2018 from the ERA5-Land, i.e. a replay of the land component of ERA5 re-analysis of the European Center for Medium-Range Weather Forecasts (ECMWF) (Hersbach et al., 2018). This database has a spatial resolution of $0.1° \times 0.1°$ (about 10 km $\times$ 10 km at the Equator). The monthly data are then projected from its original $0.1° \times 0.1°$ regular grid into the crop administrative levels to match the yield data. In detail, the gridded data have been aggregated over district or department shapes: (1) if the shape is

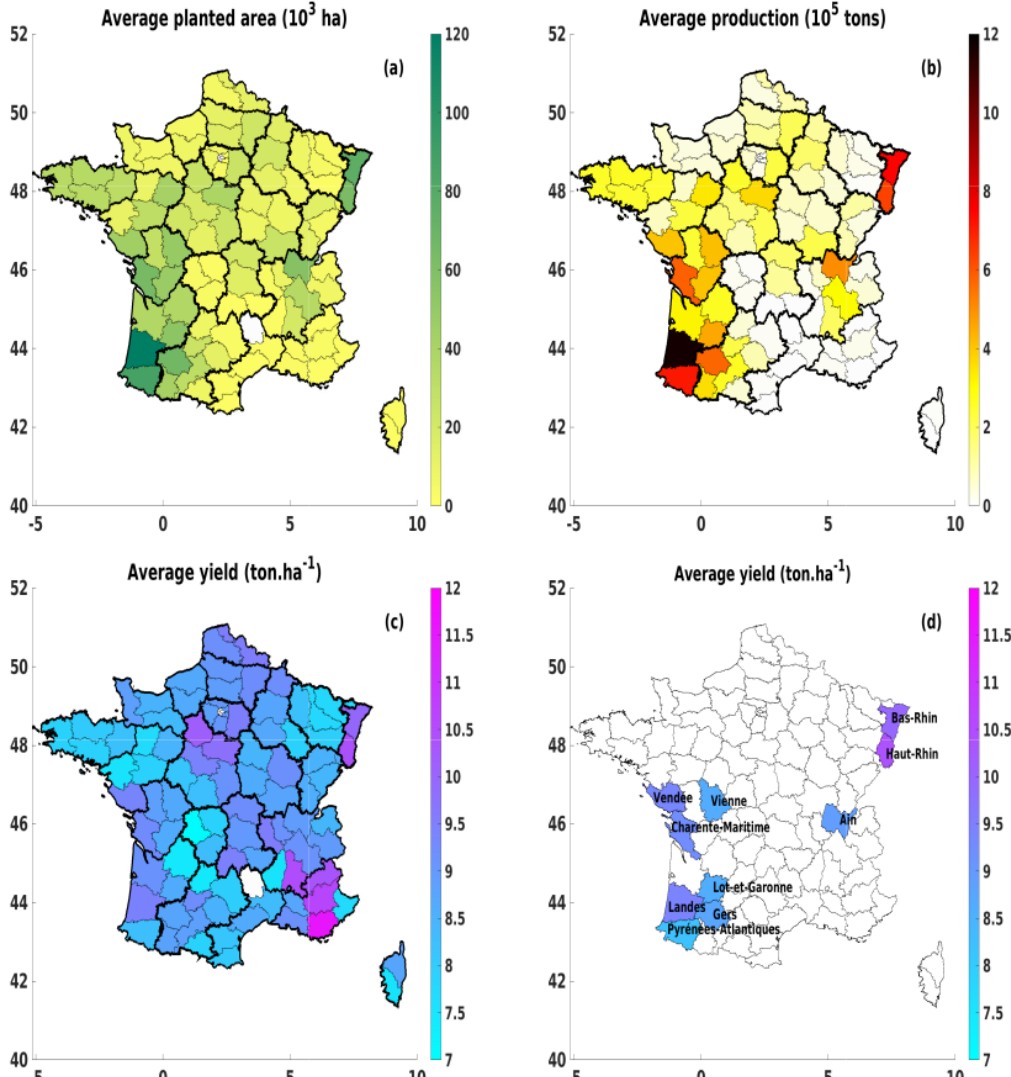

**Figure 1.** Grain maize database: (a) the average planted area (in $10^3$ ha), (b) the average production (in $10^5$ tons), (c) the average yield (in ton·ha$^{-1}$) over 96 French departments (dark lines are regions); (d) same as (c) but presenting over only ten major grain maize-producing departments. All data are averaged from 2000-2010.

smaller than the cell, the gridded value will be representative of the region; (2) if the shape includes several cells, the weather data will be averaged based on the area of cells inside the shape.

This study considers the $2 \times n$ monthly weather anomaly variables (representing P and T for $n$ months). The number of months $n$ varies for each crop:

- For Robusta coffee: we evaluated $n=19$ corresponding to the period from the bud development process to the harvest season's peak (Sect. 2.1.1). Thus, $2 \times 19$ monthly weather data (P and T from May of year $(t-1)$ to November of year $t$: $P_{May(t-1)}, \cdots, P_{Dec(t-1)}, P_{Jan(t)}, \cdots, P_{Nov(t)}$ and $T_{May(t-1)}, \cdots, T_{Dec(t-1)}, T_{Jan(t)}, \cdots, T_{Nov(t)}$) are used as potential explanatory variables for Robusta coffee yield anomalies.

- For grain maize: six months of growing period (from sowing to harvest) will be studied (Sect. 2.1.2). Thus, $n = 6$ results into $2 \times 6$ weather variables: P and T from April to September ($P_{Apr}, P_{May}, \cdots, P_{Sep}$ and $T_{Apr}, T_{May}, \cdots, T_{Sep}$).

Weather anomalies could be considered as for crop yield data. However, the climate trend of the 10 to 20 years is relatively low compared to the inter-annual variations. Thus, the long-term trend can be neglected, and the relative anomalies will be estimated based on the long-term average. This average value is computed for each of the $n$ months before the harvest time. In addition, we applied a 3-month moving average centred on the particular month (instead of the monthly data) to reduce the variability at the monthly scale. This variability would introduce instabilities in our analysis due to the short database time length. (It is actually a regularisation technique).

We also considered adding other explanatory variables (not shown), e.g. maximum and minimum temperature, and solar radiation. However, we chose not to include these variables for several reasons: (1) These variables show relatively low correlations to the crop yield anomalies; (2) They are highly correlated to P and T variables, especially for the case of Robusta coffee; (3) It will be seen in the following that considering the available yield database size, it is more reasonable to consider a limited number of explanatory variables to avoid overfitting (see more in Sect. 2.3).

## 2.2 Statistical yield models

The statistical models measure the impact of weather on crop yield anomalies, which can be denoted as: $a(t) = f_w(X)$, where $f_w$ is the parametric (or non-parametric) statistical model, $w$ stands for the model parameters, and $X$ is the set of weather inputs $\{X_i \text{ for } i = 1, 2, \cdots, n_{input}\}$. The function $f_w$ can be based on multiple statistical methods depending on the complexity of the application, for example, linear regression (Prasad et al., 2006; Kern et al., 2018; Lecerf et al., 2019), partial least-squares regression (Ceglar et al., 2016), random forest (Beillouin et al., 2020), neural network (Mathieu and Aires, 2016, 2018a), or mixed-effects (Mathieu and Aires, 2016).

In this study, two statistical models are considered:

- Linear regression (LIN) is the simplest model and the most frequently used. The relationship between the crop yield anomalies $a$ and the weather variables $X_i$ is formulated as:

$$a = \alpha_0 + \alpha_1 X_1 + \cdots + \alpha_n X_{n_{input}} + \epsilon, \tag{2}$$

where $\alpha_i$ are the regression coefficients, $\epsilon$ is the error term. Detailed description of the LIN model can be found, for example, in Dinh and Aires (2019).

- Neural Network (NN) is a non-linear statistical model. The simplest type of NN is the feedforward model (Bishop, 1995; Schmidhuber, 2015), where there is only one direction—forward—from the input nodes, through the hidden nodes and

to the output nodes. Only one hidden layer with $n_{neuron}$ neurons is considered in the architecture here. The output crop yield anomaly $a$ is modelled by the following equation:

$$a = \sum_{j=1}^{n_{neuron}} w_j \times \sigma \left( \sum_{i=1}^{n_{input}} w_{ji} X_i + b_{hidden} \right) + b_{output} \tag{3}$$

where $w$ are the weights, $b$ are the NN biases. A detailed description of the NN model (applied for impact models) is described, for instance, in Mathieu and Aires (2016).

The least-squares criterion, which minimises the discrepancies between the model predictions and observed values, is used to optimise the model during the calibration process for both LIN and NN models. It is used to obtain the coefficients $\alpha_i$ in Eq. (2) and the NN parameters $w$ in Eq. (3) during the training stage.

Two diagnostics are considered here to measure the quality of the yield anomaly estimations. (1) The Pearson's correlation COR (unitless) between the estimated $a_{est}$ and observed $a_{obs}$ yield anomalies. (2) The Root Mean Square Error is defined as: RMSE $= \sqrt{\frac{1}{n_{samp}} \sum_{i=1}^{n_{samp}} (a_{est}(i) - a_{obs}(i))^2}$. It includes systematic and random errors of the model. The RMSE unit is the same as $a(t)$; RMSE=40 represents an anomaly error of 40 %.

## 2.3 Model selection

Model selection is the process of selecting one model—among many candidate models—that best generalises (Hastie et al., 2009). This process can be applied across models of the same types with varying model hyperparameters or across different model types. Here we investigate some practically important factors of the model selection:

– **Number of inputs:** The inputs are variables that are necessary for model execution through algorithms. The inputs are selected among the potential predictors. We often have a big set of potential predictors (e.g. all-weather variables during the crop growing season), but we select only some variables from this set as the model inputs. The number of inputs defines the model complexity: the higher the number of inputs is, the more complex the model is (supposed that other factors are fixed).

– **Number of potential predictors:** The potential predictors (i.e. potential explanatory variables) here refer to all possible variables that can potentially impact the yield. Our study considers 38 weather variables for Robusta coffee and 12 variables for grain maize (Sect. 2.1), but these numbers could be much larger. For instance, in addition to selected weather variables, we could consider other variables (e.g. water deficit, soil moisture), agro-climatic indices (e.g. degree-days, free frost period (Mathieu and Aires, 2018b)). Here, we use monthly variables, but weekly or daily variables could have been considered. Therefore, establishing the list of potential predictors is not fixed in the model selection: it is a crucial modelling step preliminary to any input selection (Ambroise and McLachlan, 2002; Hastie et al., 2009).

– **Model types:** We perform the selection among two model types (presented in Sect. 2.2) with different complexity levels. For example, with $n_{input}$ inputs, a simple LIN model usually requires $(n_{input} + 1)$ parameters (Eq. (2)), while

a feedforward NN model with one hidden layer and one output requires many more parameters: $(n_{input} \times n_{neuron} + n_{neuron}) + n_{neuron} + 1$, where $n_{neuron}$ is the number of neurons in the hidden layer. A case of NN model with with varying $n_{neuron}$ will also be investigated. The number of parameters in the model is often used as a proxy for the model complexity.

## 2.4 Overfitting

When performing the model selection, it is possible to artificially fit better the training dataset. For example, increasing the model complexity can increase the model quality because a more complex model can better fit the training data. However, such a simple reasoning is dangerous: the model complexity can be too high compared to the limited information included in the training database. This limitation leads to the overfitting (or overtraining) problem, i.e. the model fits the training dataset artificially well but it cannot predict well data not present in the training dataset. Thus, an overfit model makes poor predictions and is not reliable. There is no general rule determining the model complexity based on the number of samples. An empirical tool needs to be used to check the adequacy of the model. In the following, by studying the sensitivity of the model quality to different complexity levels, we want to determine the optimal statistical crop model that truly estimates the yield anomalies as best as possible.

## 2.5 Training, validation and testing datasets

One of the main challenges in statistical inference is that the model is set up using a samples database, but it must perform well on new—previously unseen—samples. For that purpose, the overall database $\mathcal{B}$ needs to be divided into three datasets: $\mathcal{B} = \mathcal{B}_{Train} + \mathcal{B}_{Val} + \mathcal{B}_{Test}$ (Ripley, 1996):

- The **training dataset** $\mathcal{B}_{Train}$ is used to calibrate the model parameters once the model structures has been chosen.

- The **validation dataset** $\mathcal{B}_{Val}$ is a sample of data held back from the training dataset, which is used to find the best model. For instance, it helps tune the model hyper-parameters: choose the more adequate inputs (i.e. feature selection), determine the number of predictors, find the best model type (LIN, random forest, NN), determine some training choices.

- The **testing dataset** $\mathcal{B}_{Test}$ is held back from the training and the validation datasets to estimate the true model generalisation ability.

The process of partitioning $\mathcal{B}$ will be called in the following as the "folding" process. For example, the folding choice can be chosen using $\mathcal{B}_{Train} = 50\,\%$, $\mathcal{B}_{Val} = 25\,\%$, and $\mathcal{B}_{Test} = 25\,\%$.

The need for the validation dataset is often misrepresented in the literature on crop modelling. The training dataset is used to fit the parameters; the testing dataset is often used to estimate the model quality but also to choose the best model (as in the LOO approach). However, using only this testing dataset without a validation dataset brings a risk of choosing the model that best suits to this particular testing dataset. This represents a special kind of overfitting, which is not on the model calibration but on the model choice. If the database is big, many samples in the testing dataset will be representative enough; therefore, choosing

the best model based on it is acceptable. If the database is small (as often in crop modelling tasks), the model selection can be too specific for the particular samples of the testing dataset; thus, an overfitting problem can appear (Sect. 2.4). We demonstrate in the following sections that using only the testing dataset instead of the testing and validation datasets can be misleading. We avoid this difficulty by having a dataset to calibrate the model (training) and another one to choose the best model (validation). The truly independent testing dataset is then used to measure the model generalisation ability to process truly unseen data.

## 3 Measuring the quality of statistical yield models

With a limited number of samples, the training process may need every possible data point to determine model parameters (Kuhn and Johnson, 2013). It is thus impossible to keep a significant percentage of the database for the validation and the testing datasets. To choose an adequate model and avoid overfitting, a robust way to measure the generalisation ability is necessary, using as few samples as possible. Cross-validation (Allen, 1974; Stone, 1974) was introduced as an effective method for both
model selection and model assessment when having a small number of samples.

### 3.1 Traditional Leave-One-Out

The LOO method is one common type of cross-validation in which the model uses only two datasets: one to train, another to choose the model and evaluate the result. The main idea of LOO is that given $n_{samp}$ available samples in $\mathcal{B}$; the model is calibrated $n_{samp}$ times using $(n_{samp} - 1)$ samples in the training dataset $\mathcal{B}_{Train}$ (leaving one sample out). The resulting
model is then tested on the left sample ($\mathcal{B}_{Test}$). There are $n_{samp}$ testing score estimations, one for each sample. In this case, $\mathcal{B} = \mathcal{B}_{Train} + \mathcal{B}_{Test}$ and $\mathcal{B}_{Val}$ is empty. The averaging of these $n_{samp}$ testing scores is expected to be a robust assessment of the model ability to generalise on new samples. However, since no validation dataset is used to select the best model, the choice of the best model may be biased towards this testing dataset (Cawley and Talbot, 2010). The chosen model is not independent of the testing dataset, and thus, the obtained testing score is not reliable.

### 3.2 Proposed Leave-Two-Out

LOO is very useful in many cases (Kogan et al., 2013; Li et al., 2019) but as described in Sect. 2.5, it is preferable to divide the database into three partitions, rather than only two as done under LOO. In the following sections, we describe a procedure for Leave-Two-Out nested cross-validation (LTO), which can improve model selection when the number of samples is low.

#### 3.2.1 Folding scheme

For LTO, we will divide the database into three datasets: training, validation, and testing. Each time we split or partition the dataset is referred to as a "fold." Each fold divides the database $\mathcal{B}$ into a training dataset $\mathcal{B}_{Train}$ of $(n_{samp} - 2)$ samples, a validation $\mathcal{B}_{Val}$ and a testing $\mathcal{B}_{Test}$ datasets with **one sample each**. Two samples are considered out of the training dataset instead of one in the LOO procedure. This folding process is presented in Fig. 2, with the number of folds $n_{fold} = n_{samp} \times (n_{samp} - 1)$. This is why this approach is also called $k \times l$-fold cross-validation when $l = k - 1$.

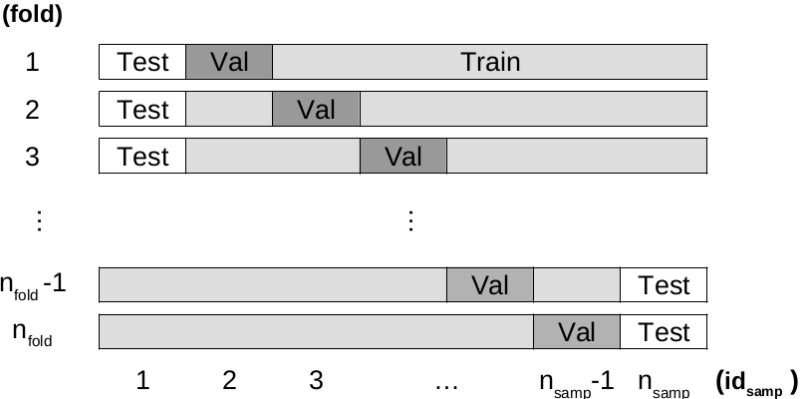

**Figure 2.** Folding strategy for the LTO procedure with $n_{fold} = n_{samp} \times (n_{samp} - 1)$ folds (corresponding to the $n_{fold}$ rows). In each fold, there are one testing, one validation, and $(n_{samp} - 2)$ training samples.

### 3.2.2 Validation and testing scores

Figure 3 illustrates how the LTO evaluation procedure is conducted. In part (a1), the number of candidate models $n_{mod}$ (represented in the horizontal axis) is defined with a fixed complexity $\lambda$ of the model. For instance, for the LIN3 model (i.e. LIN model with three inputs) with 12 potential predictors, we obtain $n_{mod} = C_{12}^3 = 220$ models. These models are used to perform the yield anomaly estimations. In the vertical axis, for each of the $n_{samp}$ choices of the testing value $id_{test} \in \{1, 2, \cdots, n_{samp}\}$, there are $(n_{samp} - 1)$ possible validation datasets, and thus training datasets. These $(n_{samp} - 1)$ training datasets correspond each to the training of the models in the horizontal axis, i.e. to fit model parameters. So $(n_{samp} - 1)$ validation and $(n_{samp} - 1)$ testing estimations are obtained for each one of the $n_{mod}$ models. The averaged validation score is used to choose the best model $bm_i$ for $i = 1, 2, \cdots, n_{samp}$; this is the role of the validation dataset.

Each choice of the testing value (each $id_{test}$) corresponds to a selected best model $bm_i$ and two distributions (i.e. Probability Density Functions (PDFs)) for $(n_{samp} - 1)$ validation errors and $(n_{samp} - 1)$ testing errors, shown in Fig. 3(a2). These two distributions result in a validation score (blue dot) and a testing score (red dot). The shape of these distributions give the average goodness-of-fit score and its variance.

Finally, the $n_{samp}$ testing choices give $n_{samp}$ validation and $n_{samp}$ testing scores that form a validation PDF in blue line, a testing PDF in red line, and thus the two scores $V_\lambda$ and $T_\lambda$ in Fig. 3(b).

A pseudo-code is provided in "Appendix A" to facilitate the implementation of the LTO procedure in any language.

### 3.2.3 Generalisation ability versus model selection

The process represented in Fig. 3 is used to obtain the validation ($V_\lambda$) and testing ($T_\lambda$) scores from the LTO approach for a given model complexity, for instance, here $\lambda$ represents the number of inputs. Each different number of inputs (different value of $\lambda$) results in different values of $V_\lambda$ and $T_\lambda$. The $V_\lambda$ and $T_\lambda$ evolution curves obtained for validation and testing RMSE values

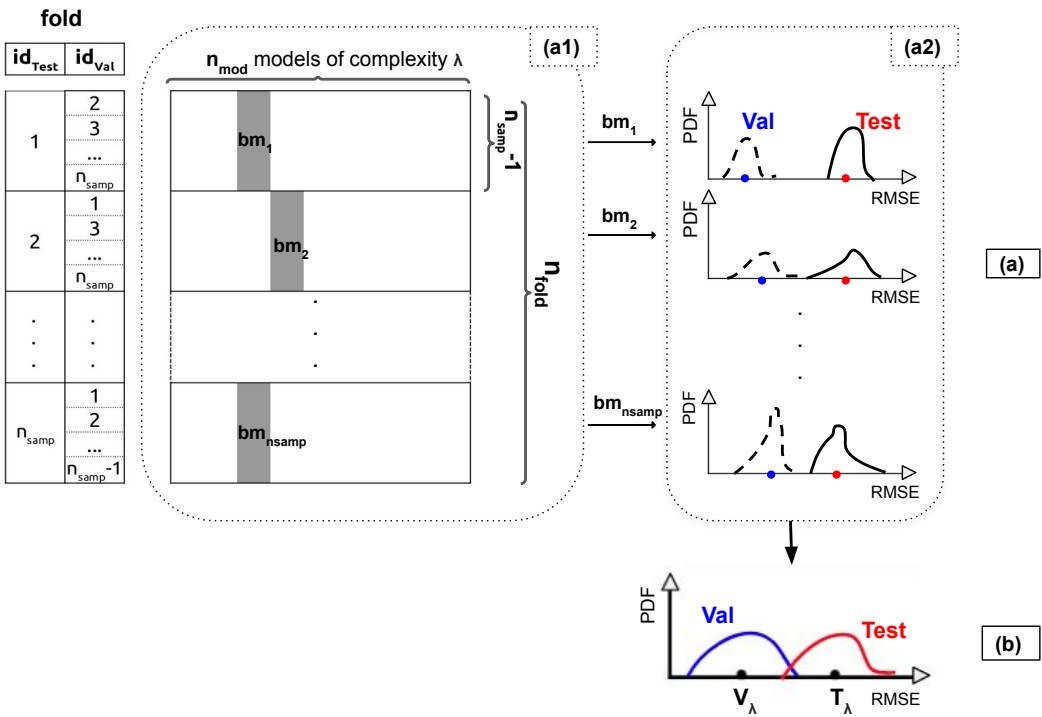

**Figure 3.** Illustration of the LTO procedure to estimate a model quality for a fixed complexity level $\lambda$ with $n_{mod}$ candidate models (horizontal axis). (a) The model errors obtained for each candidate model and each fold of the database $\mathcal{B}$ (vertical axes); (b) The obtained RMSE values for the validation and testing datasets. (See detailed description in Sect. 3.2.)

of yield anomalies for an increasing number of inputs are presented in Fig. 4. For simplicity, only validation and testing scores will be discussed since the training error almost always decrease with number of inputs. Also, the cases of underfitting are excluded in this example. When increasing the number of inputs, the validation error is smaller but the testing error is bigger; this is typical from overfitting (Sect. 2.4). In the following applications (Sect. 4 and 5), we will study these evolution curves for different models with various choices (e.g. number of inputs, number of potential predictors, model types) in order to identify the appropriate yield models for Robusta coffee and grain maize.

## 4  Robusta coffee in Cu M'gar

The first application concerns the statistical yield modelling of Robusta coffee in Cu M'gar (Dak Lak, Vietnam). The goal is to find a model that makes the most robust predictions of crop yield anomaly as a function of weather variables. We first assess several models (with varying number of inputs or number of potential predictors) to find the appropriate model choices using both LOO (Sect. 3.1) and LTO (Sect. 3.2) approaches.

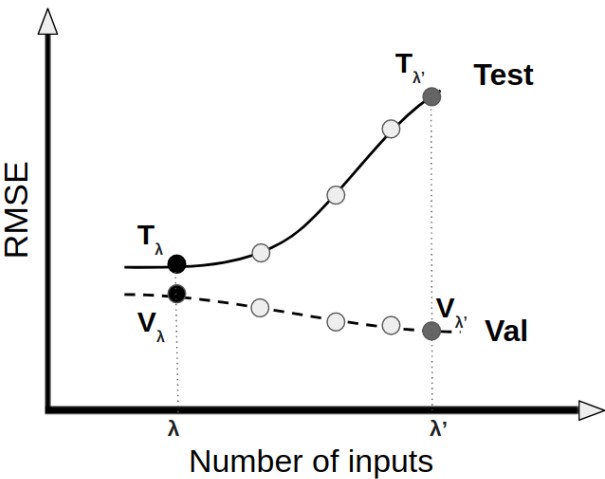

**Figure 4.** Schematic illustration of validation and testing RMSE values of predicted yield anomalies for an increasing number of inputs, obtained from the LTO procedure. For a fixed complexity level, i.e. $n_{input} = \lambda$, two RMSE values are obtained: $V_\lambda$ for validation and $T_\lambda$ for testing datasets. (The cases of underfitting are excluded in this illustration.)

### 4.1 Yield model selection

We first investigated the model choice by varying the number of inputs. In this example, the number of potential predictors is fixed to 18 ($n_{pre}$=18). The number of inputs is chosen from one to six, as shown on the horizontal axis in Fig. 5. We used the LOO and LTO procedures to compute the corresponding training, validation, and testing RMSE values. The results from LOO procedure (in Fig. 5(a)) tell us that a model with more inputs is preferable: both training and testing RMSE values decrease with the increase of the number of inputs. In the LTO case, the training and validation RMSE values decrease with the model complexity, similar to the training and testing errors in the LOO procedure. This similarity is because the LTO validation dataset has the same role as the LOO testing dataset: to find the best model! In the case of a too simplistic model, i.e. LIN model with one input, underfitting occurs as the errors are high in the training, validation, and testing datasets (shown in Fig. 5(b)). These errors decrease gradually with the number of inputs, i.e. from one to three. However, the testing errors do increase when the model has more than three inputs. The LTO procedure indicates that a simple model—with only three inputs—is optimal.

Figure 6 shows the RMSE values of the predicted Robusta coffee yield anomalies for the LIN models, with the number of potential predictors ranging from 5 to 38 (on the horizontal axis). These values are computed using the LOO and LTO procedures for the training, validation, and testing datasets. Several models have been tested; we presented here a particular example of LIN5 model, which is the linear regression model with five inputs. These inputs are selected among the considered potential predictors. For instance, for LIN5 model with six potential predictors, LOO and LTO aim at choosing five inputs among $\{P_{Nov(t-1)}, P_{Nov(t)}, T_{Mar(t)}, T_{Jan(t)}, T_{May(t)}, P_{Oct(t-1)}\}$.

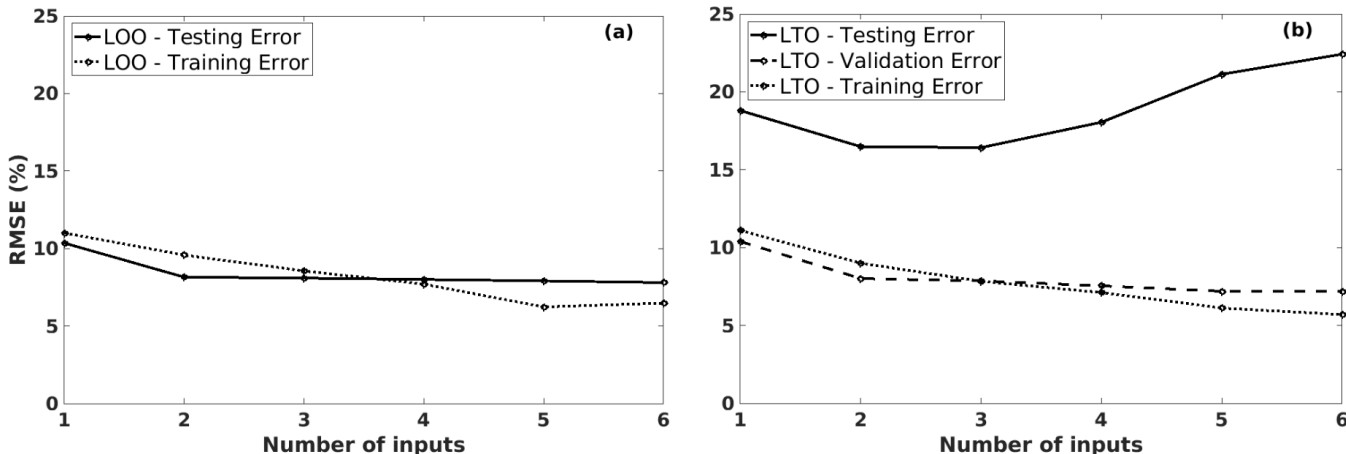

**Figure 5.** The training/validation/testing RMSE values of the predicted coffee yield anomalies, using different LIN models (with 18 potential predictors) by increasing the number of inputs, in Cu M'gar (Dak Lak, Vietnam): (a) is induced from LOO procedure, (b) is from LTO procedure.

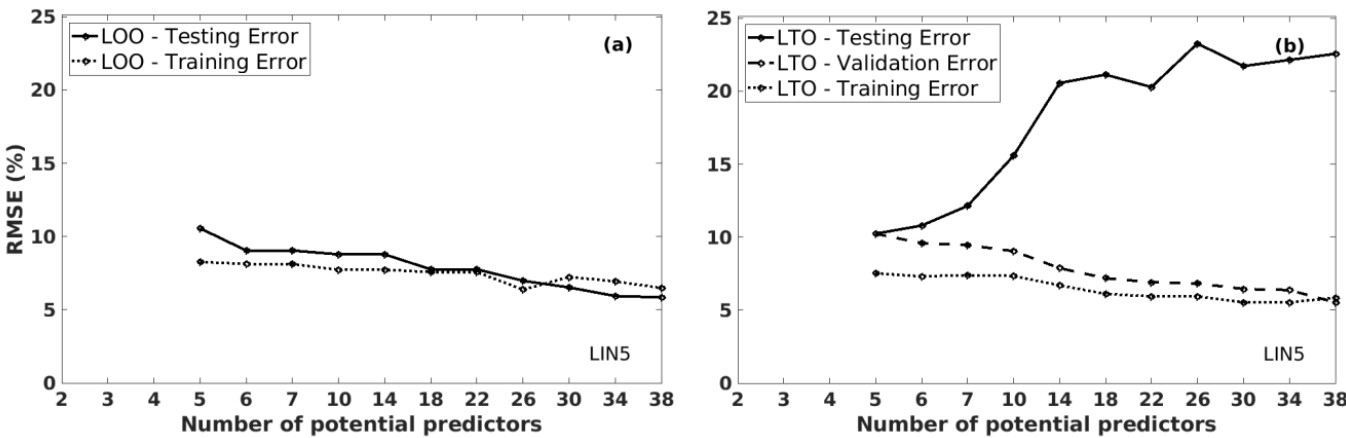

**Figure 6.** The training/validation/testing RMSE values of the predicted coffee yield anomalies, using LIN5 models by increasing the number of potential predictors, in Cu M'gar (Dak Lak, Vietnam): (a) is induced from LOO procedure, (b) is from LTO procedure.

The LOO procedure suggests that the more potential predictors the models have, the better results are. Both training and testing RMSE values decrease gradually (Fig. 6(a)) with the increase of the number of potential predictors for LIN5 models. On the other hand, the same behaviour is observed for the LTO procedure in Figs. 6(b): the testing errors show an opposite trend to the training/validation errors and gradually increase with the number of potential predictors. The LTO procedure indicates that a simpler model with fewer potential predictors is more adequate. This conclusion makes sense since it is inappropriate to use a very complex model (as the LOO model choice) when having a limited sample.

The LOO procedure is actually misleading because it could encourage us to choose a model that overfits the data: the same testing dataset is used to choose the best model and to assess the generalisation ability. If the modellers select the best model based on information from the LTO procedure, they are less likely to choose an overfit model. As in this case, they choose the model on the validation dataset and assess its generalisation score on an independent testing dataset.

In short, considering the limited information in the available database—that is used to train, select the model, and evaluate its quality—it is not possible to use more than a very simple and limited model. Therefore, for this 19-sample coffee yield modelling case, using a simple LIN model is better than a complex one.

## 4.2 Yield anomaly estimation

The previous section showed that the LTO procedure allows us to choose a reasonable model, simple enough, with fewer inputs and potential predictors. Thus, the crop yield estimations of the LTO method will be assessed here to see how good the selected model (LIN3 model with three predictors) is. The final model includes $\{P_{Nov(t-1)}, P_{Nov(t)}, T_{Mar(t)}\}$ and these selected variables coincide with the key phenological phases of Robusta coffee. For example, there is the need for a dry period for the buds to develop into dormancy at the end of the development stage, i.e. Nov(t-1) (Schuch et al., 1992). Therefore, $P_{Nov(t-1)}$ impacts directly the buds, thus the potential yield. Similarly, the fruit maturation stage (Nov(t)) benefits from weather conditions with less precipitation. At the beginning of the fruit development period (Mar(t)), too low temperature slows maturation rate to the detriment of yield, while a higher temperature is beneficial (Wintgens, 2004).

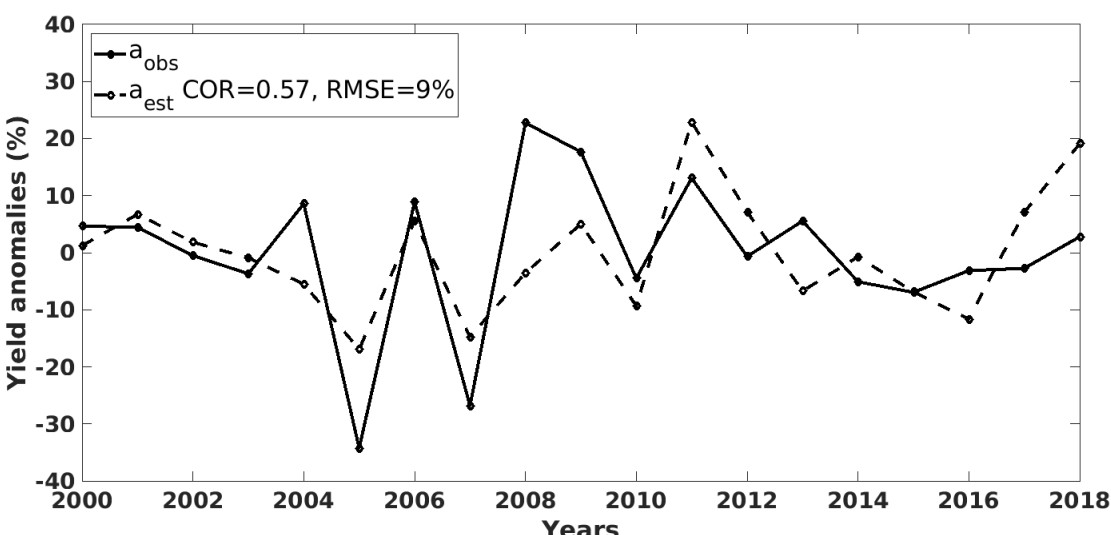

**Figure 7.** The observed (solid line) and LTO estimated (dashed line) coffee yield anomalies time series in Cu M'gar (Dak Lak, Vietnam).

Figure 7 presents the estimated yield anomalies time series for Robusta coffee in Cu M'gar from 2000 to 2018. The estimation ($a_{est}$ in the dashed line) describes quite well the observations ($a_{obs}$ in the solid line) with a correlation of 0.57. With

precipitation and temperature variables, the selected model is able to identify many extreme years (e.g. 2005-2009, 2010, 2011) or a decreasing trend from 2011 to 2015. Also, the correlation score means that the model can explain more than 30 % ($0.57^2$) of the variation in coffee yield anomalies. This value is reasonable as the weather is among several factors (e.g. prices, sociotechnical factors, managerial decisions) affecting coffee yield (Miao et al., 2016; KC et al., 2020; Liliane and Charles, 2020). It is possible to apply the resulting statistical crop yield model to future climate simulations and then study the impact of climate change on coffee (Bunn et al., 2015; Craparo et al., 2015a; Läderach et al., 2017).

## 5 Grain maize over France

This application considers several aspects of grain maize over France. First, the sensitivity of the forecasting quality to the model selection is studied, using the LOO and LTO approaches, over Bas-Rhin and Landes—the major grain maize-producing departments and all 96 departments in France. Then, the forecasting scores are investigated over ten major grain maize-producing departments.

### 5.1 Yield model selection - Focus on Bas-Rhin and Landes

In this section, we describe how we selected the most appropriate statistical model for grain maize using 22 years of yield data. This test is done over Bas-Rhin and Landes (i.e. two major grain maize-producing departments in France). As shown in Sect. 4.1, the LOO approach can be misleading and cause the analyst to select overly complex models. Thus, we focus here on the LTO results for different models with various selections: number of inputs, model types, number of neurons in hidden layer, and number of potential predictors.

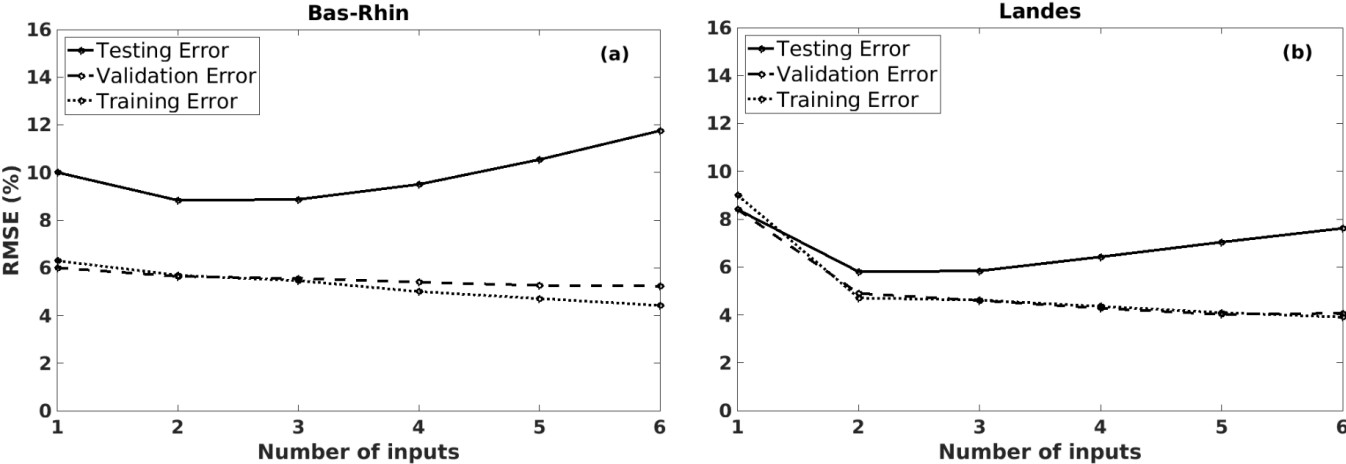

**Figure 8.** The training/validation/testing RMSE values of the predicted grain maize yield anomalies, using different LIN models (with 12 potential predictors) by increasing the number of inputs, in (a) Bas-Rhin and (b) Landes.

Similar to Robusta coffee case (Sect. 4.1), we fixed the number of potential predictors $n_{pre} = 12$ and gradually increased the number of inputs from one to six in the horizontal axis of Fig. 8. Again, in both Bas-Rhin and Landes examples, underfitting occurs when models are too simple, for example, with one input. With a higher number of inputs, the LTO procedure shows a similar behaviour as previous examples (Sect. 4.1): the validation/training errors decrease gradually, while the testing errors show an opposite trend. This behaviour suggests that a simple model (e.g. LIN3 for both Bas-Rhin and Landes) is more adequate.

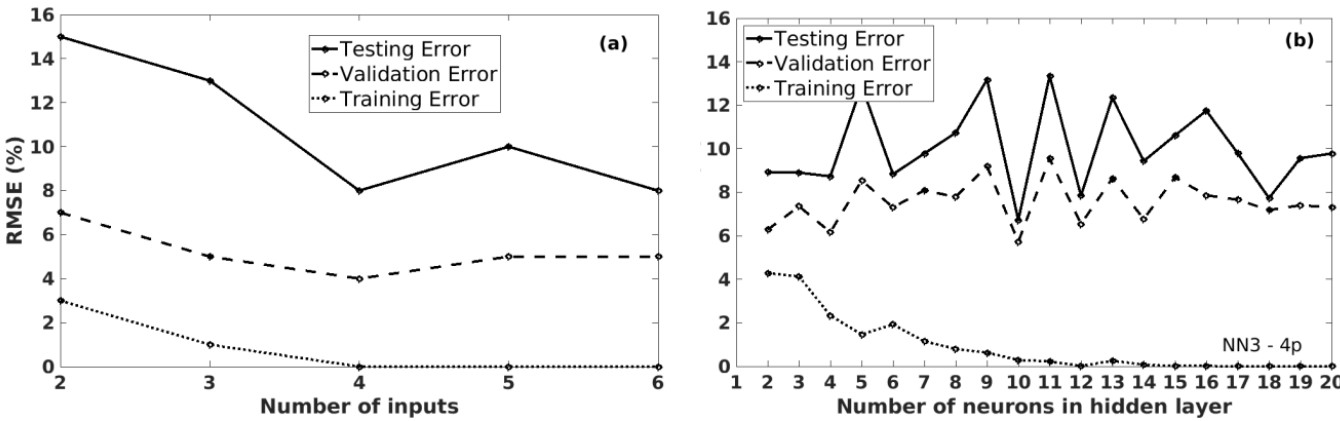

**Figure 9.** The training/validation/testing RMSE values of the predicted grain maize yield anomalies, using different NN models with various choices, in Bas-Rhin (France): (a) NN models (with $n_{pre} = 12$ and $n_{neuron} = 7$) by increasing the number of inputs, (b) NN3 models (with $n_{pre} = 4$) by increasing the number of neurons in the hidden layer.

More complex models were tested in Fig. 9: (a) NN models (with 12 potential predictors and seven neurons in the hidden layer) by increasing the number of inputs, (b) NN3 models (with four potential predictors) by increasing the number of neurons in the hidden layer. The impact of overfitting (Sect. 2.3) is noticeable when the model is too complex. For instance, in both cases (Fig. 9), the training errors get smaller—close to 0—for more inputs or more neurons in the hidden layer, as expected. However, the testing and validation errors show large fluctuations when increasing the model complexity. These fluctuations imply that the model is overfitted, and thus, random error or noise appear. Similar results (not shown) were obtained for NN3 models with $n$ potential predictors, where $n = 3, 7, 12$. Thus, the NN models are unreliable due to the limited number of samples to train a non-linear model.

We also tested other examples with LIN3 and NN3 models (Fig. 10) to illustrate the cases where model types and number of potential predictors affect the model quality. Figure 10 describes the RMSE values of the predicted grain maize yield anomalies for three datasets (training, validation, and testing) of the LTO procedure. The results of LIN3 models are presented in Fig. 10(a), and NN3 models (with seven neurons in the hidden layer) are in Fig. 10(b), with a different number of potential predictors ranging from 3 to 12 in the horizontal axis. The same behaviours are observed: the validation/training errors decrease, while the testing errors increase with the number of potential predictors. Also, the NN3 models show much higher testing and

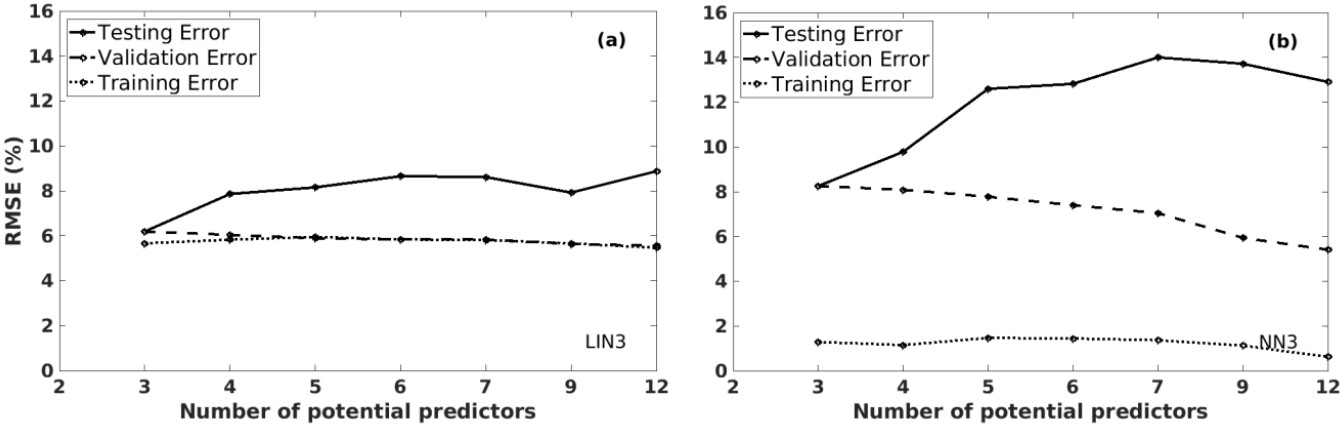

**Figure 10.** The training/validation/testing RMSE values of the predicted grain maize yield anomalies, using different models by increasing the number of potential predictors, in Bas-Rhin (France): (a) LIN3 and (b) NN3 (with $n_{neuron} = 7$) models.

validation RMSE values compared to the LIN3 models. Again, we can conclude—in this grain maize application—that a

simpler model will be more beneficial than the complex one.

**5.2   Reliability model assessment**

In this section, a statistical yield model is applied first over 96 French departments to assess the true model quality. Then, we will focus on ten major departments to assess how the selected models perform for yield anomaly predictions.

Figure 11 shows the true testing RMSE maps of predicted grain maize yield anomalies in France. Here, the testing errors

induced from the LTO procedure are used on the models chosen by the LOO and LTO approaches. In other words, both methods (LOO and LTO) can be considered to identify optimal crop models, but only the LTO method is used (as a reliable tool) to estimate the model generalisation ability. For example, when considering only LIN3 models, LOO chooses models with 12 potential predictors, while LTO chooses three. From these choices, the true model generalisation scores (i.e. testing errors) are estimated using the LTO approach, shown in the RMSE maps of Figs. 11(a1) and (b1). Another example focuses on

LIN5 models (presented in Figs. 11(a2) and (b2)). The true errors obtained from the LOO choice are clearly higher than those from the LTO choice for LIN3 models. For instance, the testing RMSE values range from 10 % to 18 % in many departments in Fig. 11(a1), while these values are often lower than 10 % in Fig. 11(b1). This difference shows that the LOO approach under-estimates these true errors, as seen in Fig. 11(a1). Thus, the model choice of the LOO approach is misleading. For more complex models like LIN5 models—that is preferred by the LOO choice—in the second row of Fig. 11, the higher errors are

observed, especially for LOO model errors of many northern departments with up to 22 % of RMSE (Fig. 11(a2)). This grain maize application confirms the benefit of LTO to select and assess the true quality of statistical yield models, while LOO is misleading by under-estimating the true errors of its selected models. A simple LIN3 model with three potential predictors is adequate for this application considering the limited amount of data.

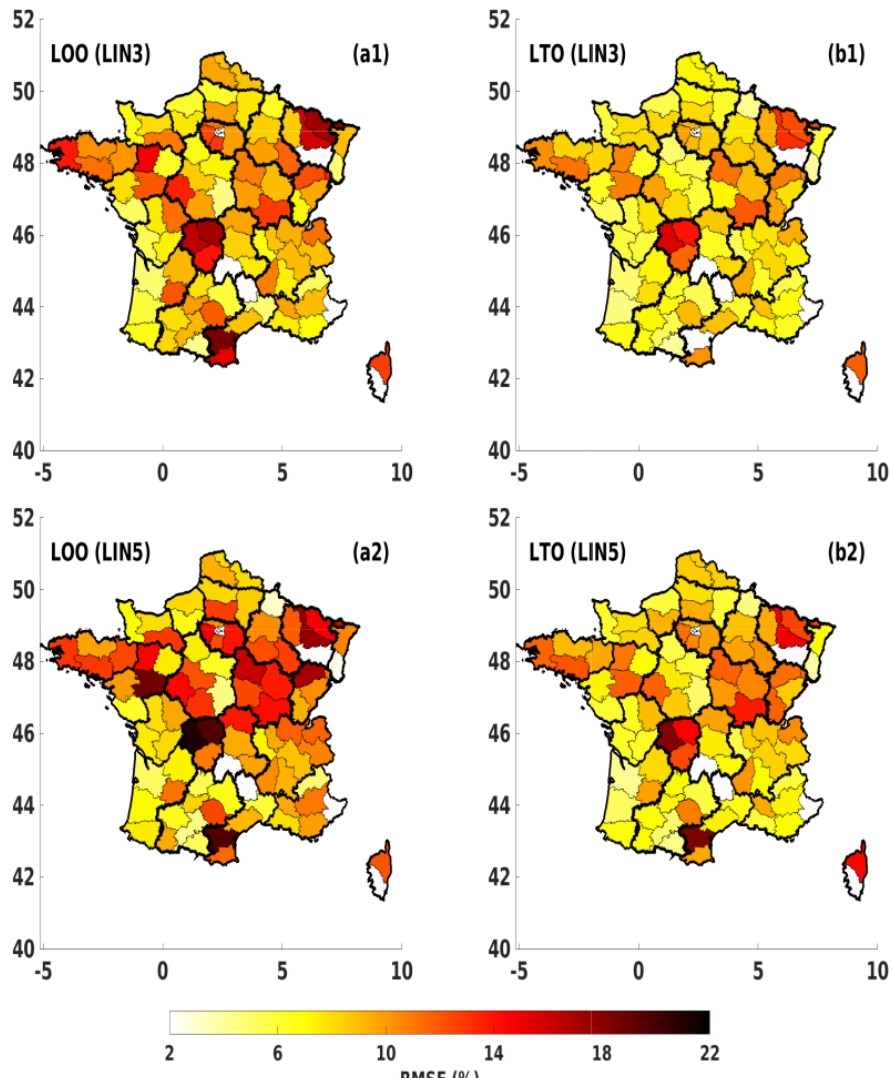

**Figure 11.** The true testing RMSE maps of predicted grain maize yield anomalies in France for LOO (first column) and LTO (second column) approaches, induced from two LIN models with a different number of inputs: LIN3 (first row) and LIN5 (second row).

We now analyse how good the LTO testing estimations are compared to the observations over ten major grain maize-producing departments (as shown in Fig. 1(d)). Figure 12 presents the boxplots of residuals for these departments, which are the differences between the observed and estimated yield anomalies (Residual=$a_{obs} - a_{est}$ in %). The medians of the residuals lie near zero. It means that the selected models can predict the yield anomalies with acceptable coverage and precision. Although there are some extreme values (Lot-et-Garonne) and some outliers, the interquartile, which ranges from about -8 % to 8 %, shows slight differences between the observations and estimations over study departments.

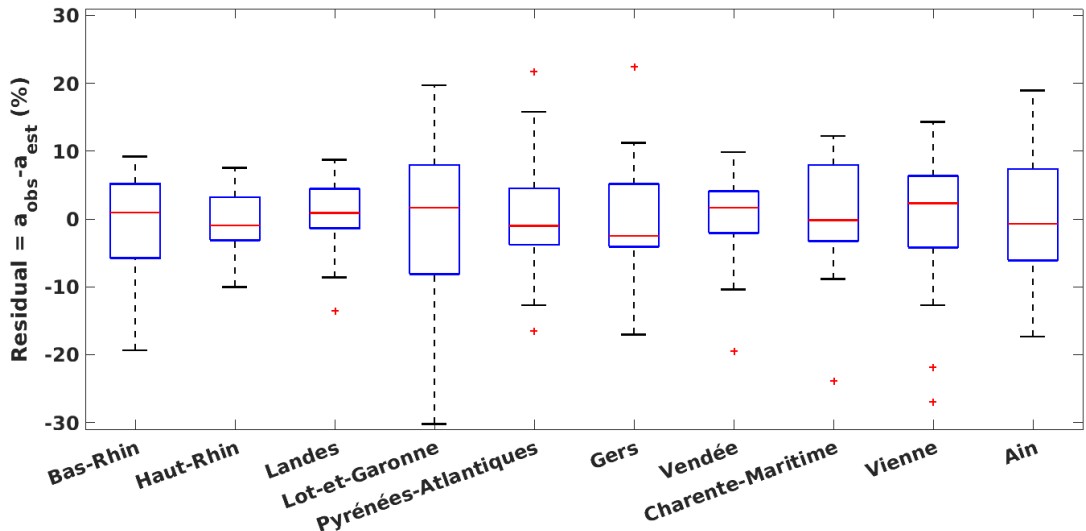

**Figure 12.** Boxplots of residuals (the difference between the observed and estimated yield anomalies) for ten major grain maize-producing departments: red horizontal bars are medians, boxes show the 25th-75th percentiles, error bars depict the minimum and maximum values, and red + signs are suspected outliers.

## 5.3 Seasonal yield forecasting

The LTO approach is helpful for selecting an adequate model with better forecasting. Here, the model chosen by the LTO procedure is tested for seasonal forecasting, from the sowing time (April) to the forecasting months (i.e. from June to September): all-weather variables (including P and T) from April to June can be selected for the June forecasting. Table 1 represents the correlations between the observed and estimated yield anomalies of the forecasts from June to September. The quality of the seasonal forecasting models gradually increases when approaching the harvest because more information is provided. With the weather information at the beginning of the season (April, May, and June), the June forecasting model obtains an average correlation of 0.35 between the observations and estimations. This score is significantly improved when adding information of July (correlation of 0.51). This improvement means that the weather in July strongly influences grain maize yields. The improvement from July to August is much less than from June to July, with an average increase of 0.01 and 0.16, respectively. No information is added in the September forecasting model since it coincides with the harvest time. In other words, the final model should consider only variables from April to August. As in our case, statistical model selects $\{T_{Jul}, P_{May}, P_{Apr}\}$ as the final inputs for grain maize in the eatern region (Bas-Rhin, Haut-Rhin); $\{T_{Jul}, P_{Jul}, T_{Apr}\}$ for the southern region (e.g. Landes, Pyrénées-Atlantiques, Gers); and $\{P_{Jul}, P_{Apr}, P_{Jun}$ or $T_{Jun}\}$ for the central part (Vendée, Charente-Maritime, Vienne). It is reasonable to have different inputs for different regions (or even departments) due to their distinct environmental conditions. In general, weather variables in July—the flowering period—are among the most influential variables. During this time, a high temperature affects the photosynthesis process, thus reducing the potential yield; in contrast, positive precipitation anomalies

are preferable (Ceglar et al., 2016; Mathieu and Aires, 2018b). Precipitations in April and May also show significant impacts on grain maize as a water deficit during this vegetative stage decreases plant height (Çakir, 2004).

| Departments | Forecasting months | | | |
|---|---|---|---|---|
| | June | July | August | September |
| Bas-Rhin | 0.46 | 0.47 | 0.47 | 0.47 |
| Haut-Rhin | 0.35 | 0.53 | 0.53 | 0.53 |
| Landes | 0.63 | 0.64 | 0.66 | 0.67 |
| Lot-et-Garonne | 0.02 | 0.22 | 0.22 | 0.29 |
| Pyrénées-Atlantiques | 0.34 | 0.60 | 0.60 | 0.60 |
| Gers | 0.33 | 0.61 | 0.60 | 0.43 |
| Vendée | 0.63 | 0.63 | 0.63 | 0.63 |
| Charente-Maritime | 0.21 | 0.52 | 0.53 | 0.62 |
| Vienne | 0.39 | 0.40 | 0.40 | 0.40 |
| Ain | 0.17 | 0.52 | 0.52 | 0.52 |
| **Average** | **0.35** | **0.51** | **0.52** | **0.52** |

**Table 1.** The correlation between the observed and estimated yield anomalies for different forecasting months (from June to September), over ten major grain maize-producing departments.

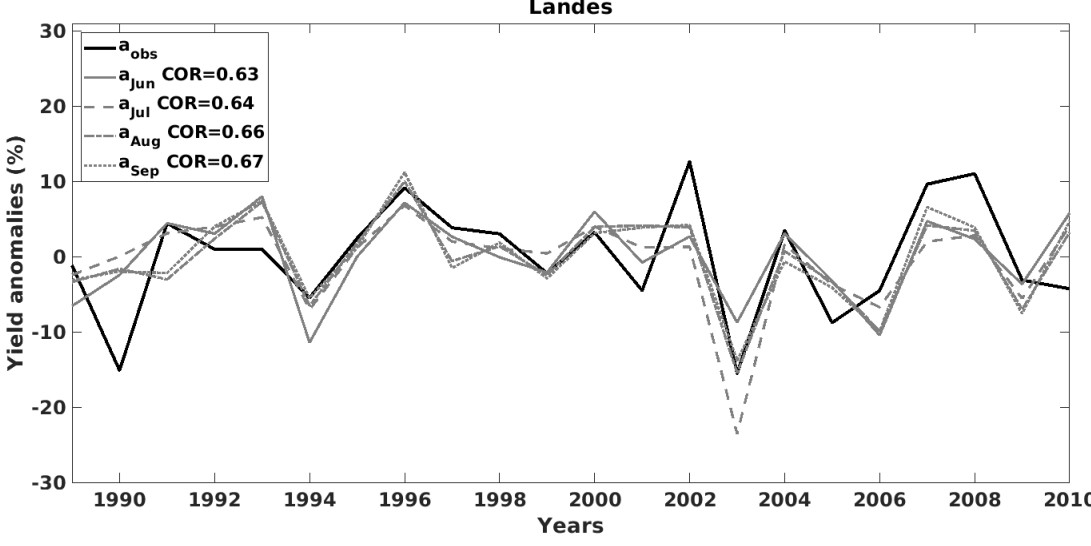

**Figure 13.** The observed ($a_{obs}$) and the estimated yield anomalies time series, for different forecasting months from June to September (e.g. $a_{Jun}$ means June forecasting), for grain maize in Landes (France).

In addition, Fig. 13 shows time series plots of the yield anomaly observations and estimations for different forecasting months in Landes (France). In this case, the June forecasting results show a high correlation with the observed yield anomalies (0.63). This score slightly increases when approaching the harvest. It also indicates that the weather can explain more than 40 % ($0.67^2 = 44.89$ %) of variations in grain maize yield anomalies in this region, which is in line with other crop studies (Ray et al., 2015; Ceglar et al., 2017). However, the forecasting models cannot predict all the extremes (e.g. negative yield anomaly in 1990) that are probably influenced by the climate extremes (Hawkins et al., 2013; Ceglar et al., 2016). The statistical models could be improved by adding the indices that focus on extreme weather events.

## 6  Conclusions and perspectives

Crop yield modelling is very useful in agriculture as it can help increase the yield, improve the production quality, and minimise the impact of adverse conditions. Statistical models are among the most used approaches with many advantages. The main difficulty in this context is the limitation of the available crop databases to calibrate such statistical models. Applications typically rely on only two or three decades of data. This small sample size issue directly impacts the complexity level that can be used in the statistical model: a model too complex cannot be fit with such limited data, and assessing the true model quality is also challenging. In practice, statistical inference requires three datasets: one for calibrating the model, a second one for choosing the right model (or tuning the model hyper-parameters), and another for assessing the true model generalisation skills. Dividing a very small database into such three datasets is very difficult.

The LOO method has been used as a cross-validation tool to calibrate, select, and assess competing models (Kogan et al., 2013; Zhao et al., 2018). It was shown in this paper that LOO can be misleading because it uses only one dataset to choose the best model and estimate its generalisation skills simultaneously. This is a true problem as LOO is one of the main statistical tools to obtain crop yield models. This study proposes a particular form of nested cross-validation approach that we call a LTO method. This method helps select the best model by using two datasets that are independent of the training dataset: first the validation dataset is used to select the best model form or complexity, and then the test dataset is used to independently assess the model performance. In our case studies of crop yield prediction, LTO shows that only very simple models can be used when the database is limited in size. The LTO implementation proposed here is very general and can be applied to any statistical crop modelling problem when the number of samples is small and a large number of potential predictors are available.

Two applications have been considered. The first one concerns the coffee yield modelling over a major Robusta coffee-producing district in Vietnam. It was shown that considering the available historical yield record, the best statistical model can explain about 30 % of the coffee yield anomaly variability. The remaining variability is rather large, and may be explained by non-climatic factors (e.g. prices, sociotechnical factors, managerial decisions, or political and social context). It could also come from climate; however, the model would require more detailed variables (e.g. at a daily scale) or more samples to go into deeper details of the climate-crop yield relationship. In addition, explaining a third of the coffee yield variability is in line with the literature (Ray et al., 2015; Craparo et al., 2015b). LTO was able to identify the suitable model trained on the historical record and estimate the true model ability to predict yield on independent years. The final model includes {$\mathrm{P}_{Nov(t-1)}$, $\mathrm{P}_{Nov(t)}$,

$T_{Mar(t)}$}, which corresponds to the key phenological phases of Robusta coffee: the end of the bud development, the fruit maturation, and the beginning of the fruit development, respectively.

The second application is related to grain maize yield in France. The LTO was used here to choose between simple linear models and more complex neural network models. Our findings also show that LOO was misleading in overestimating the testing scores. LTO indicated that a simple linear model is preferable because it has a lower testing error. This approach can also be helpful in seasonal forecasting applications (during the growing and the beginning of harvest seasons). In this application, the weather can explain more than 40 % of the yield anomaly variability, which is similar to that reported in the literature (Ray et al., 2015; Ceglar et al., 2017). This score can vary depending on study regions because some regions are more sensitive to the climate than others. Generally, grain maize yield anomalies are mainly influenced by weather variables during the flowering period (July) and the early season (April).

In the future, the mixed-effects model can be considered instead of a straightforward statistical model. This approach—which intends to use samples in several regions (e.g. gathering samples into groups) to compensate for the lack of historical data—could help us obtain more complex crop models (Mathieu and Aires, 2016). Such a mixed-effect could benefit from the LTO scheme. In terms of applications, the crop models that we derived here could be used on climate simulations (from an ensemble of climate models for the next 50 years) to investigate the crop yield sensitivity to climate change. In addition, by using a similar approach presented here, other crops will be investigated, for instance, over France (Ceglar et al., 2016; Schauberger et al., 2018; Ceglar et al., 2020), over Europe (Ceglar et al., 2017; Lecerf et al., 2019) or globally (Bunn et al., 2015). Furthermore, these types of statistical crop models can be used to refine the potential adaptation and mitigation strategies. For instance, it is expected that the climate runs could help us identify the change in optimality for the crop culture in the world.

*Code availability.* The Matlab code used to run an example of the leave-two-out method is available at the following Zenodo link for the revision process of GMD: https://zenodo.org/record/5159363 (Anh and Filipe, 2021).

*Data availability.* The coffee data were provided by Vietnam's General Statistics Office (GSO) for the 2000-2018 period. These data are available from GSO on reasonable request. For any inquiries, please send an email to banbientap@gso.gov.vn. The data on French grain maize (and other French crops) are available at http://agreste.agriculture.gouv.fr from 1989 on. In addition, the weather data, i.e. ERA5-Land data, can be downloaded from https://cds.climate.copernicus.eu (last access: 22 Apr 2021).

# Appendix A: Appendix

```
n_samp = number of samples; %years
n_pre = number of potential predictors;
n_mod = number of models;
n_fold = number of folds of the dataset;
Score(2,n_fold,n_mod); %representing RMSE or COR; 2 for [Test,Val];
```

```
      bm = best model ϵ {1,···,$n_{mod}$};
      %Step 1: Build scores for each fold, each model
for inp = 1 to $n_{fold}$
          %Define the folding process
          Test = 1 sample ϵ {1,···,$n_{samp}$};
          Val = 1 sample ϵ {1,···,$n_{samp}$} - Test;
          Learn = {1,···,$n_{samp}$} - Test - Val;
for imod = 1 to $n_{mod}$
              %Train models
              model = train(model, Learn);
              Score(1,inp,imod) = RMSE(model, Test);
              Score(2,inp,imod) = RMSE(model, Val);
end
      end
      %Step 2: Choose best model for all folds; estimate its score
      for isamp = 1 to $n_{samp}$
          $Mean_{Val}$ = mean(Score(2,$n_{fold}$\{isamp\},:)); %(1,1,$n_{mod}$)
ibm(isamp) = $\underset{i}{argmin}$($Mean_{Val}$);
          $Score_{Test}$(isamp) = mean(Score(1,$n_{fold}$\{isamp\},ibm(isamp))); Test score
          $Score_{Val}$(isamp) = mean(Score(2,$n_{fold}$\{isamp\},ibm(isamp))); Val score
      end
      $FinalScore_{Test}$ = mean($Score_{Test}$)
$FinalScore_{Val}$ = mean($Score_{Val}$)
```

*Author contributions.* All authors conceptualized the research and formulated the model. LADT implemented the model in Matlab and analyzed the output with FA. All authors contributed to writing the paper.

*Competing interests.* The authors declare that they have no conflict of interest.

*Acknowledgements.* We are grateful to Matthew Heberger for his useful suggestions for improving this manuscript. We thank the editor and
495 the reviewers for their valuable comments and suggestions on our manuscript.

This work is a part of Lan Anh's Ph.D., which benefited from the French state aid managed by the ANR under the "Investissements d'avenir" program with the reference ANR-16-CONV-0003; and the Australian Centre for International Agricultural Research, conducted as part of activities for the project "Enhancing smallholder livelihoods in the Central Highlands of Vietnam through improving the sustainability of coffee and black pepper farming systems and value chains" (AGB/2018/175).

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
