# Peer review of "Nested leave-two-out cross-validation for the optimal crop yield model selection"

_Geoscientific Model Development, 2021_

## Author Comment (AC1)

We thank the referee for his/her helpful comments and suggestions.

*Here we would like to give our quick response concerning some comments of the referee. A point-to-point response will be available later when we correct and update the manuscript as a revised version.*

We totally agree with the referee that the problem should be presented in a better way. To improve the manuscript, we plan to add one/more sections about the characteristic of the model, crop information, and/or weather-yield analysis. These factors had been already considered, but we did not show them in the manuscript, so we will add them in the revised version.

We agree with the referee that the (fundamental) analysis is much more complicated than the present weather-yield impact model. In reality, many factors (e.g., soil characteristics, the spillover effects from previous seasons, or extreme events) could affect the yield. For instance, the extremes during the growing season can largely influence the yield of that year (Beillouin et al., 2020; Mathieu and Aires 2018; Vogel et al., 2019). However, in practice, this is not possible for a statistical model. In fact, the crop database is, most of the time, very limited in time (often about 10 to 20 years). This means that there are not enough samples to calibrate a very complex statistical model with many input factors and a description of their interaction.

Actually, the main objective of this paper is to introduce the leave-two-out technique that measures, in a robust way, the true capacity of a statistical crop model. This technique told us in the two studied crops here that we simply cannot introduce more input parameters in the statistical model; this would be misleading and wrong.

You are right; it is a bit misleading when we say we can explain 30% of the yield variance with climate. We mean that considering the historical yield record, we can only set up a statistical model that can explain 30%. This is a lower estimate, and climate could explain more than that, but to go into deeper details of the plant physiology, we would require many more samples. Please note that we are extremely rigorous in our statistical modelling practice (this is why we introduced this leave-two-out method) and that many other studies are not so rigorous and claim they can explain a larger part of the variance. We think this is not honest, and it comes from the over-fitting process. We were probably not clear enough in the first version of the paper, and we hope that the new version will clarify the overall meaning and strategy of our analysis.

Concerning the resolution of climate data, the 0.1° x 0.1° resolution data should be considered an adequate input for this type of statistical model over different administrative levels (i.e., district level and department level). Also, the data are compatible with what can be obtained from

climate models (e.g., CMIP6 (Eyring et al., 2016)) and thus is adapted to the climate change impact study that we want to perform later on.

Finally, we also plan to change the title: "Using the leave-two-out method to determine **the optimal complexity** of the statistical crop model" to be more explicit about the target of the study. This is not to optimise a very sophistical model that would explain all the crop physiology, but instead to find the optimal complexity of the model in a context of a very short time-record database.

**References:**

Beillouin D, Schauberger B, Bastos A, Ciais P, Makowski D. Impact of extreme weather conditions on European crop production in 2018. Philos Trans R Soc Lond B Biol Sci. 2020 Oct 26;375(1810):20190510. doi: 10.1098/rstb.2019.0510. Epub 2020 Sep 7. PMID: 32892735; PMCID: PMC7485097.

Eyring, V., Bony, S., Meehl, G. A., Senior, C. A., Stevens, B., Stouffer, R. J., and Taylor, K. E. (2016). Overview of the Coupled Model Intercomparison Project Phase 6 (CMIP6) experimental design and organization. Geoscientific Model Development, 9(5):1937–1958.

Mathieu, J. A., & Aires, F. (2018). Using neural network classifier approach for statistically forecasting extreme corn yield losses in Eastern United States. Earth and Space Science, 5, 622–639. https://doi.org/10.1029/2017EA000343

Vogel, E., Donat, M. G., Alexander, L. V., Meinshausen, M., Ray, D. K., Karoly, D., Meinshausen, N., & Frieler, K. (2019). The effects of climate extremes on global agricultural yields. Environmental Research Letters, 14(5), [054010]. https://doi.org/10.1088/1748-9326/ab154b

---

## Author Comment (AC2)

**Authors response**

*Title: Using the leave-two-out method to determine the optimal statistical crop model*
*Author(s): Thi Lan Anh Dinh and Filipe Aires*
*MS No.: gmd-2021-218*
*MS type: Methods for assessment of models*

\---------
Dear referees,

We thank the three referees for their highly constructive comments and suggestions. Our responses (in blue) are listed below. The updated manuscript (i.e., tracked changes version) is also included underneath our responses.

We hope that our corrections will make the manuscript clearer and pleasing to the referees and other readers.

Thanks & best regards,
Lan Anh
\---------

**Referee comment 1 :**

The authors show the value of a two out of sample approach for statistical modelling. The formal approach is convincingly motivated and presented.

Thank you for your appreciation.

The substance of the modelling, however, is completely inadequately presented. One would expect a brief characteristic of the modelled crops and their cropping specifics (annual, perennial; sensitive phases for weather dependence), a characterisation of the climate at the site considered, information about the crop relevant weather variability in the periods considered, a motivation of the predictors, and the margins for determining the predictors, a discussion of model errors.

We agree that the problem should be presented in a better way. To improve the manuscript, we added more information in Sect.2.1 about the characteristics of the model, crop information, characteristics of study areas (climate, agricultural practices), and crop-weather relation. These factors had been already considered, but we did not show them in the manuscript, so we added them in the updated version. We believe that the substance of the modelling is now adequately introduced.

The superficiality of the problem analysis becomes particularly clear in the classification of the model quality:

'It was shown that monthly mean precipitation and temperature could explain more than 30 % of the coffee yield anomaly variability. The 70 % remaining variability is due to non-climatic factors (agricultural practices, diseases, or political and social context). '

This conclusion presupposed that the authors had fully explained the weather-related yield variability with their approaches. However, this cannot be assumed given the selection of climate variables, their coarse resolution, the static view on the underlying processes, the negligence of their interplay, the linearity restrictions applied, the exclusion of spillover effects from previous seasons and of soil characteristics.

We agree with the referee that the (fundamental) problem is much more complex than our weather-yield impact model. In reality, many factors (e.g., soil characteristics, the spillover effects from previous seasons, or extreme events) could affect the yield. For instance, the extremes during the growing season can largely influence the yield of that year (Beillouin et al., 2020; Mathieu and Aires 2018; Vogel et al., 2019). However, in practice, it is not possible for a statistical model to take into account such complexity. In fact, the crop database is, most of the time, very limited in time (often about 10 to 20 years). This means that there are not enough samples to calibrate a very complex statistical model with many input factors and a description of their interaction.

Actually, the main objective of this paper is to introduce the leave-two-out technique that measures, in a robust way, the true capacity of a statistical crop model. For the two studied crops,

this technique shows us that we simply cannot introduce more input parameters in the statistical model; this would be misleading and wrong.

You are right; it is a bit misleading when we say that weather can explain more than 30% of the yield anomaly variance, it could explain more than that if enough samples were available for the calibration of the impact model. We mean that considering the available historical yield record, we can only set up a statistical model that can explain 30 %. This is a lower estimate, and climate could explain more than that, but we would require many more samples to go into deeper details of the plant physiology. Please note that we are extremely rigorous in our statistical modelling practice (this is why we introduced this leave-two-out method) and that many other studies are not so rigorous and artificially claim they can explain a more significant part of the variance. We were probably not clear enough in the first version of the paper and we hope that the new version will clarify the overall meaning and strategy of our analysis.

Concerning the resolution of climate data, we believe that the 0.1° x 0.1° resolution data (~10 x 10 km in the Equator) should be considered as an adequate input for this type of statistical model over different administrative levels (i.e., district level and department level). Please note that monthly-mean values are smooth in nature and that increasing resolution would not necessarily offer more details. Furthemore, the considered districts in Vietnam cover an area of about 822 $km^2$ and the average area of a department is about 6 000 $km^2$. Also, the data are compatible with what can be obtained from climate models (e.g., CMIP6 (Eyring et al., 2016)) and thus are adapted to the climate change impact study that we want to perform later on.

Nevertheless, the presented approach shows which formal possibilities exist to obtain a 'best' model without an in-depth examination of the object to be modelled. This is quite interesting for practical statistical modelling. However, the examples presented here (maize, coffee) should be motivated much more comprehensively and classified in a more differentiated way. This does not mean that the authors have to make a comprehensive modelling claim. But they should be able to justify and classify the range of the model configurations considered in their modelling attempt.
As we mentioned above, we added descriptions about the details of the two considering crops and models. Once again, we want to highlight the point that data scarcity is the main problem of statistical models for such applications. You are right, we should introduce better that potentially, a more complex model could be considered. However due to the limited amount of data available to constrain such a model, we have no alternative than to use a simpler one. This is very clear in the abstract and the introduction.

The general quality of the manuscript is currently at the border between 'fair' and 'poor', but has potential for moving to the better direction when the authors motivate more explicitly their modelling approach for the selected climates and crops.

We were not clear enough about the general context of this modelling, and we hope that our corrections improve the manuscript clarity.

We changed the title: "Using the **nested** leave-two-out **cross-validation** method to determine the **optimal complexity** of the statistical crop model" to be more explicit about the target of the study. This is not to optimise a very sophisticated model that would explain all the crop physiology but instead to find the model's optimal complexity for a very short time-record database.

Although the term "leave-two-out"/ LTO will be used throughout the manuscript for simplicity purposes, we mentioned the "**nested** leave-two-out **cross-validation**" in the title to avoid confusion for the reader at first glance.

**References:**

Beillouin D, Schauberger B, Bastos A, Ciais P, Makowski D. Impact of extreme weather conditions on European crop production in 2018. Philos Trans R Soc Lond B Biol Sci. 2020 Oct 26;375(1810):20190510. doi: 10.1098/rstb.2019.0510. Epub 2020 Sep 7. PMID: 32892735; PMCID: PMC7485097.

Eyring, V., Bony, S., Meehl, G. A., Senior, C. A., Stevens, B., Stouffer, R. J., and Taylor, K. E. (2016). Overview of the Coupled Model Intercomparison Project Phase 6 (CMIP6) experimental design and organization. Geoscientific Model Development, 9(5):1937–1958.

Mathieu, J. A., & Aires, F. (2018). Using neural network classifier approach for statistically forecasting extreme corn yield losses in Eastern United States. Earth and Space Science, 5, 622–639. https://doi.org/10.1029/2017EA000343

Vogel, E., Donat, M. G., Alexander, L. V., Meinshausen, M., Ray, D. K., Karoly, D., Meinshausen, N., & Frieler, K. (2019). The effects of climate extremes on global agricultural yields. Environmental Research Letters, 14(5), [054010]. https://doi.org/10.1088/1748-9326/ab154b

**Referee comment 2 :**

**General comments**

This paper addresses the very important question of model complexity. How can one best choose the level of model complexity when the objective is to minimize error of prediction for out of sample cases. The paper presents two practical cases (prediction of coffee yield in Viet Nam, prediction of maize yield in France) and two prediction methods (linear regression, artificial neural network). The main objectives of the paper are to explain the leave two out cross validation approach (LTO) as a method of evaluating and choosing between models of different levels of complexity, and to compare it with the leave one out cross validation approach (LOO).

We thank you a lot for your constructive comments.

The presentation of LTO is useful and interesting, but I have some issues with the way the linear regression examples are formulated. (I do not comment on the use of neural networks, with which I am not very familiar).

Our response to your specific comments and technical corrections are listed below.

**Specific comments**

The authors consider models with a fixed number of explanatory variables and take as the measure of model complexity, the number of potential explanatory variables from which the explanatory variables in the model are chosen. This is not usually the way the problem is formulated, at least in linear regression. In general, the list of potential explanatory variables is fixed, and the question is how many and which to include in the final model. Then the larger the number of explanatory variables chosen, the more complex the model. Comparing LTO and LOO as a function of the number of potential explanatory variables may then not be very relevant to the problem of determining model complexity for a linear model. A more relevant question would be: How do LTO and LOO compare, when the number of potential explanatory variables is fixed, and the objective is to choose the best ones to include in the regression model?

You are right, establishing the optimal number of inputs in a fixed list of potential predictors is often the problem a modeller is facing. We actually consider this case in Sect.4.1: when the number of potential predictors is fixed to 30 and then we increase the number of inputs from 2 to 7 (shown in Fig.6(b)).

As the main target of the paper is to determine the optimal complexity of the statistical crop model, we want to investigate different factors that control model complexity. Model complexity is a wide field of research (e.g., Vapnik–Chervonenkis dimension), in this manuscript we consider several parameters controlling this complexity: the number of potential predictors, the number of inputs, and the two model types (LIN versus NN).

The question of how long should be the list of potential predictors is also a true concern for the modeller: The crop experts have the tendency to increase this list a lot because many parameters can have an impact on the crop. Furthermore, the modeller needs to choose the temporal resolution of the predictors (we used monthly, but weekly or daily values could have been chosen. In Mathieu and Aires (2018), a list of six climate indices {$T_{Jul}$, $T_{Aug}$, $P_{Jul}$, $T_{Jun}$, $T_{May}$, $P_{Aug}$} have been proposed by many experts to perform statistical crop yield modelling. When doing a "plan of experience", the way statisticians define the list of potential predictors is therefore a strong decision, and we show here that it has a significant impact on the model performance. For instance, Fig.5(b1) shows that selecting three inputs out of 38 potential predictors gives a much higher LTO RMSE testing error than selecting three inputs out of 18 potential predictors.

In general, more inputs make the model more complex. Our study shows that the larger number of potential predictors also defines the model complexity. Thus, the number of potential predictors should be considered as an essential factor when doing any statistical analysis. But any parameter controlling the model complexity should be considered: It is actually the goal of this study to propose a tool to optimise these factors.

We added more information in the manuscript (e.g., Sect. 2.3 and Sect. 4.1) to better explain our performed experiences and our main targets. We hope that these changes can accommodate your concerns.

In addition, though LOO can be used for model selection (choosing best explanatory variables) in linear regression, there are other approaches which are probably more common, such as forward regression, stepwise regression, the Akaike Information Criterion etc. How does LTO compare with other methods of model selection?

Yes, there are other common approaches for model selection. We actually used the forward selection (Sect. 2.3.1). These tools are not incompatible with the LTO. It is actually necessary to have a way to test for model errors in the forward selection: LOO is often used (Mark and Goldberg, 2001) for that purpose. Here our main message is that we must use LTO instead of LOO to determine the optimal model, including the choice of forward regression.

**Technical corrections**

L138 Problem with English
We changed the sentence to:
"This leads to the overfitting (or overtraining) problem, i.e., the model fits the training dataset artificially well but it cannot predict well data not present in the training dataset. Thus, using this type of model is not reliable."

L145 Should be "to choose"
Yes. We corrected it.

L163-164 I don't understand the sentence
We changed the sentence to:
"It will be seen in the following that using only the testing dataset instead of the testing and validation datasets can be misleading."

Section 3, introduction. It seems to me that much of this is said elsewhere. Maybe combine with section 2.3?
Thanks for the comment. We shortened the introduction of Sect. 3 and added some ideas into Sect. 2.3.

Fig 3. I find the portrayal of distribution functions confusing. First of all, in the figure the distribution functions are continuous, while in practice they are necessarily discrete. Secondly, as far as I can tell, the distribution functions are totally irrelevant. Only the average error is of interest.
Thanks for the comment. Here, the distribution functions intend to better illustrate the full process of the folding scheme: the validation and testing scores are first obtained for all folds of $id_{test}$; these scores give the blue and red distributions; after that, the final average scores are computed from these two distributions.
The PDF functions can also give information about the uncertainties of the final score. For instance, if the spread of the blue distribution is small, it means that our estimation of the validation score is characterised by a low uncertainty.
Also, we think there is no need to have detailed discrete values in a scheme like this. We synthesise them as continuous values.
Therefore, we would like to keep this figure as it is.

Section 3.2.2. Talking of a "testing dataset" is a bit confusing (is this the full set of testing data, which is identical to the available data, or is this a single value for each fold). Perhaps refer to the "testing value" or "testing datum" when talking about a single fold.
We used the "testing value" to avoid confusion.

L266 "implemented" is probably the wrong word.
We changed it to "exploited"

L362 "request" is probably the wrong word
It should be "prefer", but we removed this sentence due to the repetition of the previous sentence (at the beginning of this paragraph).

L366 I don't understand this sentence.
We made changes for this sentence and the following sentences:
"... .The first one concerns the coffee yield modelling over a major Robusta coffee-producing district in Vietnam. It was shown that considering the available historical yield record, we can only set up a statistical model that explains about 30 % of the coffee yield anomaly variability."

**Referee comment 3 :**

The authors discuss an important aspect of statistical analysis with limited data availability and exemplify the shortcomings of the more frequently used 'leave one out' cross validation compared to the 'leave two out'. Two case studies are used, an annual crop in Europe and a perennial crop in Vietnam. Two different statistical approaches are used, linear regression and neural network. The two case studies convincingly demonstrate how to select the best model for forecasting crop yield with limited data availability, while considering the problem of overfitting. In general the study is well designed and clearly communicated, although substantial English revision is required.

Thank you very much for your appreciation on this work. The manuscript has been improved. We hope that this updated version accommodates all your concerns.

**Some general observations:**

- Line 132: the model architecture is explained as consisting of i) number of potential predictors, and ii) the number of inputs. Please specify clearer what the difference is between predictors and inputs. The authors also mention model types: Some models require more parameters to estimate even with equal numbers of predictors. This should be described in clearer terms.

  We added more explanations in this section, i.e., Sect. 2.3.1:

  "Various factors control the complexity level of a statistical model: the model architecture (the number of potential predictors, the number of inputs, the number of parameters or the model types (e.g., linear or non-linear)) or the training process (e.g., the number of epochs in NN or the loss function).In theory, it is challenging to define the exact definition of a model complexity: even the number of parameters in the models is only a proxy because a model with a low number of parameters can be highly complex, e.g., Vapnik–Chervonenkis dimension (Hastie et al., 2009). This study thus investigates some of the factors that control part of the model complexity.

  **Number of inputs**

  The inputs are variables that are necessary for model execution through algorithms. The inputs are selected among the potential predictors. We often have a big set of potential predictors (e.g., all-weather variables during the crop growing season), but we select only some variables from this set as the model inputs. The number of inputs defines the model complexity: the higher the number of inputs is, the more complex the model is (supposed that other factors are fixed).

  **Number of potential predictors**

  The potential predictors here refer to all possible variables that can potentially impact the yield. Our study considers 38 weather variables for Robusta coffee and 12 variables for grain maize (Sect. 2.1), but these numbers could be much larger. For instance, in addition to selected weather variables, we could consider other variables (e.g., water deficit, soil moisture), agro-climatic

indices (e.g., degree-days, free frost period (Mathieu and Aires, 2018b)). Here, we use monthly variables, but weekly or daily variables could have been considered. Therefore, establishing the list of potential predictors is not fixed: it is a crucial modelling step. The following sections (Sect. 4.1 and 5.1) will show that the number of potential predictors drives the model complexity: having too many potential predictors is dangerous, in particular, if the tools are not right.

**Model types**

Model complexity can be shown in two model types that we presented in Sec. 2.2. For example, with $ninput$ inputs, a simple LIN model requires ($ninput$ + 1) parameters (Eq. (2)), while a feedforward NN model with one hidden layer and one output requires much more parameters: ($ninput \times nneuron + nneuron$ ) + $nneuron$ + 1, where n neuron is the number of neurons in the hidden layer."

**Reference:**
Hastie, T., Tibshirani, R., and Friedman, J.: The elements of statistical learning: data mining, inference and prediction, Springer, http://www-stat.stanford.edu/~tibs/ElemStatLearn/, 2009.

- The value of using LIN3 and LIN5 are not clear to me, which is related to the previous comment: the authors explain them as linear regression models with three and five inputs, respectively. Could you explain what the value is of this experiment? What added insights do we gain when comparing LIN3 and LIN5? Do we not already get all the insights from adding the number of potential predictors to LIN3?

  In this part (Sect.4.1), we would like to test different model complexity levels by changing several factors: the number of potential predictors, the number of inputs, and the two model types (LIN, NN). Various models have been tested to resolve the problems of the number of potential predictors, but only LIN3 and LIN5 are shown (together with the LIN models in Fig.6(b)). Also, LIN5 is chosen here as it illustrates very well the robustness of the LTO method compared to the LOO: the LOO training/testing and LTO training/validation RMSE values follow an evident decreasing tendency, while this tendency is not so clear in the LIN3 example. On the other hand, LIN3 and LIN5 also are examples of model complexity defined by the number of inputs. With a more complex model (LIN5), the observed behaviour of training/validation/testing RMSE values is much stronger than a simpler model (LIN3).

  We added some changes in this section to better explain the problem.

- There is quite a bit of redundancy – the authors explain the problem several times, thereby repeating themselves. For example, section 3 (lines 168-181) already has been elaborated in previous sections. More concise presentation of the problem and methods would benefit the manuscript.

Yes. We totally agree about this. We shortened some parts of the manuscript and combined them into other sections. You can trace these changes in the tracked changes manuscript.

- Even though the main aim of the study is to compare the LOO and LTO, presenting the chosen predictors and final model would be appreciated. Any reader familiar with the case study crops will be interested to understand what climate descriptors were tested and selected in the final model. This transparency is further necessary for making the study reproducible.

Thanks for this useful suggestion. We added more information about the chosen predictors and final model (Sect. 4.2, Sect. 5.3).

For instance, in Sect. 4.2:

[revised manuscript text omitted]

---

## Author Response (AR2)

**General comments**

This is a revision. The paper addresses the very important question of model complexity. How can one best choose the level of model complexity when the objective is to minimize error of prediction for out of sample cases. The paper presents two practical cases (prediction of coffee yield in Viet Nam, prediction of maize yield in France) and two prediction methods (linear regression, artificial neural network). The main objectives of the paper are to explain the leave two out cross validation approach (LTO) as a method of evaluating and choosing between models of different levels of complexity, and to compare it with the leave one out cross validation approach (LOO). The presentation of LTO is useful and interesting. The review of the original manuscript pointed out some important problems with the specific cases used to illustrate LTO. These problems have not been adequately addressed in the revision.

*We thank the referee for his/her comments. These comments have been carefully considered, and modifications have been made accordingly.*

**Specific comments**

The major problem is in the definition of model complexity. The linear models in this study have a fixed number of explanatory variables (3 for LIN3, 5 for LIN5), chosen from n candidate explanatory variables, where increasing values of n are tested. The main measure of model complexity in the paper is the number of potential explanatory variables. As pointed out in the review of the original manuscript, this is not a usual definition of model complexity. Normally, it is the number of explanatory variables in the model that is the measure of model complexity. Not only is the number of potential explanatory variables not the usual measure of model complexity, it in fact is a very problematic measure of model complexity. Using this measure of model complexity, one could have two identical models and conclude that one is much more complex than the other, if the explanatory variables were chosen from among a larger pool of candidate explanatory variables. That is, model complexity would no longer be determined by the model itself, but also by the exact history of how the model was developed. I think that to illustrate LTO as a way of choosing the best level of model complexity, the authors need to use a more accepted measure of complexity (i.e. the number of explanatory variables in the model, for a fixed set of potential explanatory variables).

*We agree that the number of potential explanatory variables (i.e., potential predictors) is not a usual definition of model complexity. However, as described in our manuscript (Sect. 2.3), the list of potential predictors is not fixed, and thus establishing this number is a crucial modelling step. Especially, in agricultural applications, **with a very limited number of samples**, it is inappropriate to consider a large number of predictors. This*

*large number of predictors issue was also identified in previous studies (Ambroise and McLachlan, 2002; Hastie et al., 2009).*

*In addition to the number of potential predictors, we did test several (accepted) measures of complexity: increasing the number of explanatory variables (i.e., inputs) for a fixed set of potential predictors) or considering a more complex neural networks model instead of a linear regression one.*

*In the revised version, we showed more results of the common measures of model complexity: the number of inputs, model types, the number of neurons in the hidden layer, as shown in Sects. 4.1 and 5.1. Also, we introduced the number of potential predictors as one factor that drives the model selection and model quality. We also used the term "model selection" instead of "model complexity" to avoid confusion for the reader and better present the problem.*

*References:*

*Ambroise, C. and McLachlan, G. J.: Selection bias in gene extraction on the basis of microarray gene-expression data, Proceedings of the National Academy of Sciences, 99, 6562–6566, https://doi.org/10.1073/pnas.102102699, 2002.*

*Hastie, T., Tibshirani, R., and Friedman, J.: Model Assessment and Selection, in: The elements of statistical learning: data mining, inference and prediction, pp. 219–260, Springer, 2009.*

---

## Author Response (AR3)

The manuscript provides an important contribution to improve current statistical crop modelling practices and clearly illustrates potential pitfalls of most common approaches. The revised manuscript is well structured and the content well presented, however, it would be of great value if the English language could be revised to facilitate a smooth reading.

Thank you for your appreciation. We have made changes to improve the manuscript. We also asked a native English scientist to proofread the manuscript. We hope that our improvement will make the manuscript clearer now.

**Anonymous referee #4**

**General comments:**

This publication presents a highly relevant validation technique for statistical crop models, i.e. a possibility to select the input variables and validate the model independently of the testing data set, which improves the robustness of the model. The LTO validation is presented in two case studies. Results show that the LTO validation leads to more robust results and enables a more realistically assessment of the forecasting performance. Because rigorous validation remains rare in the statistical crop modelling community, this paper is of high interest to other scientists. Even though the proposed approach has not often been applied yet, it is also not a new approach. Laudien et al. (2020 and 2022) and Meroni et al. (2021) present examples in which an independent variable selection has been applied to forecast crop yields. However, the explicit comparison of the influence of a different number of input variables (either inputs or potential predictors) on the model performance is – to our best knowledge - new and interesting for a wider audience. The paper is well-structured and has a clear, but partly colloquial language.

We thank the referee for his/her highly constructive comments and suggestions. Please find below our responses (in blue) to each of your comments.

**Major comments:**

- The comparison of the selection of the best number of predictors and inputs between LOO and LTO does not include the results for fewer than 3 variables. As the RMSE for the validation and the testing increases with the number of inputs, the question arises whether the optimal value is even lower than 3. Also, the paper title suggests that the optimal number of inputs is found. The risk of too simplistic models is not explored in the paper, which would be an interesting addition to the presented results.

Thanks for this constructive comment. In fact, in Figs. 5 and 8, we compared different LIN models, in which the number of inputs increases from two to six.
We now improve the comparison by adding the LIN1 model (i.e linear model with only one input) in both Figs. 5 and 8. Corresponding comments on these two figures are also added:

Section 4.1: "... In the case of a too simplistic model, i.e. LIN model with one input, underfitting occurs as the errors are high in the training, validation, and testing datasets (shown in Fig. 5(b)). These errors decrease gradually with the number of inputs, i.e. from one to three. However, the testing errors do increase when the model has more than three inputs. The LTO procedure indicates that a simple model---with only three inputs---is optimal."

Section 5.1.: "... Again, in both Bas-Rhin and Landes examples, underfitting occurs when models are too simple, for example, with one input. With a higher number of inputs, the LTO procedure shows a similar behaviour as previous examples (Sect. 4.1): the validation/training errors decrease gradually, while the testing errors show an opposite trend. …"

- Laudien et al. (2020): "Robustly forecasting maize yields in Tanzania based on climatic predictors"; Meroni et al. (2021): "Yield forecasting with machine learning and small data: What gains for grains?"; Laudien et al. (2022): "A forecast of staple crop production in Burkina Faso to enable early warnings of shortages in domestic food availability" provide examples of an independent variable selection in a statistical crop model to forecast yields. Whereas Laudien et al. call it "level 2 LOOCV", Meroni et al. (2021) also call it nested oos validation. The statement that the proposed LTO approach has never been used before (Line 50-51) is therefore not correct. Please rephrase this sentence.
Thanks for the comment. We rephrased the sentence and added your suggested references.
"We found very few applications of this approach in the literature on statistical crop modelling (Laudien et al., 2020, Meroni et al., 2021, Laudien et al., 2022)."

- As the authors state, the number of input variables, the number of potential input variables and the model type influence the robustness of the model/forecast and the inclusion of all of these aspects in the study is relevant. The terms model selection, model complexity or "finding the best model" are often used to describe the selection of inputs, potential inputs and model type even though these terms encompass also other aspects, as pointed out in section 2.3. These terms therefore do not adequately describe what the authors are examining, which leads to confusion. Also, the terms are used ambiguously, e.g. in Line 266 it says "Here, the model complexity is considered as a representative example of model selection." The paper would benefit from a clearer language concerning what is actually investigated, i.e. either input selection, potential input selection or selection of the model type.
In this study, we considered all three factors: the inputs, the potential inputs, and the model types. It is true that we did not clear enough in the manuscript. We rephrased Sect. 2.3 to better introduce the problem and factors that we investigated. Sect. 3.2.3 is also rephrased and it shows only one

example of input selection instead of using the confusing term "model complexity". We hope that all the changes made especially in Sect. 2.3 and Sect. 3.2.3 will resolve this comment.

- The reference to Dinh et al. (2022) is made about 10 times in the paper – also as a way to justify some methodological decisions – even though this paper is not yet published and has the same first author. Please consider finding alternative literature sources.
We added several alternative literature sources to replace this reference.

**Minor comments:**
- L5: It is not impossible to split the data set into 3 in statistical modelling. Please rephrase.
We changed it to: "Splitting the overall database into three datasets is often impossible in crop yield modelling due to the limited number of samples."

- L33-34: "not an easy task" is a rather subjective statement. Sentence should be rephrased.
We changed it to: "Splitting a small number of samples into three datasets is not easy."

- L 43: This statement should be supported by references.
We added related references.

- L 46: Instead of "from" – it should be "for".
Change made.

- L74, 76, 79: Do you mean reproductive stage?
For coffee, this stage is called the "productive stage".
(please see, for instance, Valeriano et al., 2018: Estimation of Coffee Yield from Gridded Weather)

- L87: Please provide a reference.
We added a related reference of Mathieu and Aires, 2016: Statistical weather-impact models: An application of neural networks and mixed effects for corn production over the United States.

- L99: Please reformulate the phrase. "change in time" is misleading. Do you mean changes in the timing of phenological stages?
We changed the sentence to: "Although the sowing time varies for different regions (Olesen et al., 2012), the average growing season of French grain maize ranges from April to September …"

- L115, L197: There is no comma after "i.e.". Please, also check at other parts of the paper.
Thanks. Change made.

- L116: It should be "at" the equator, not "in" the equator.
Change made.

- L117-118: How has the data being matched to administrative levels? This should be specified, as it influences the results.

We added more details to explain this: "In detail, the gridded data have been aggregated over district or department shapes: (1) if the shape is smaller than the cell, the gridded value will be representative of the region; (2) if the shape includes several cells, the weather data will be averaged based on the area of cells inside the shape."

- L148: The error term in the regression equation is missing.

We added this term.

- L179: The regression does not necessarily require an intercept (In case the dependent variable was demeaned beforehand, the intercept is no longer needed.). Therefore, it is not always n_input +1. Please rephrase, e.g. the LIN model "usually" requires n_input +1 inputs.

We corrected the sentence.

- L189-192: This statement does not belong to the section on Methods. Findings are presented later in the paper and should not already be mentioned at this point.

We removed this statement as the idea is mentioned in the Results and Conclusions.

- L166-67: Please rephrase. Also many other factors can influence the model performance, not only complexity and potential input variables.

As mentioned earlier, we rephrased the whole section (Sect. 2.3) to better introduce the problem.
"Model selection is the process of selecting one model—among many candidate models—that best generalises (Hastie et al., 2009). This process can be applied across models of the same types with varying model hyperparameters or across different model types. Here we investigate some practically important factors of the model selection: … "

- L219: This is "often" the case in crop modelling studies, however there are also studies with big samples (e.g. Schauberger et al. (2022): French crop yield, area and production data for ten
staple crops from 1900 to 2018 at county resolution, Lobell (2008): Prioritizing Climate Change Adaptation Needs for Food Security in 2030 or Renard, Tilmann (2019): National food production stabilized by crop diversity).

We rephrased the sentence: "If the database is small (as often in crop modelling tasks), the model selection can be too specific for the particular samples of the testing dataset …"

- L276: Applicability of a model is not only defined by its skill – please rephrase.

We rephrased the whole sentence to: "The goal is to find a model that makes the most robust predictions of crop yield anomaly as a function of weather variables."

- L296: It should be "models" not "model".

Change made.

- L300: Robust statistical models can also be based on smaller samples than 19. Please make the sentence more general by e.g. saying: "when having a limited sample."
Change made.

- L306f: It is not illusionary to model complex weather-yield relations with a sample of 19 observations - many papers show that it is possible. The choice of input variables should also account for more complex weather-yield relations (i.e. only studying monthly mean temperature or precipitation sum might not be sufficient). Rather refrain from this statement.
We removed this statement.

- L312 and L430: Do you mean key phenological phases in plant development by moments of coffee?
We changed "key moments" to "key phenological phases".

- L312-316: This is a very interesting discussion as it explains why the selected variables potentially show a good performance in the model. However, this should be supported by literature.
Thanks for the comment. We added related references to this paragraph.

- L322: Weather is only one factor among other factors. However, the examples are not well-chosen, i.e. by omitting the yield trend you deliberately omit the influence of e.g. agricultural practices (e.g. irrigation) that usually only change gradually over time. Also, one could argue that diseases are indirectly covered in statistical models. Please, refer to literature at this point to support your examples.
Thanks for your comment. We changed the sentence and added several supporting references (Miao et al., 2016: Responsiveness of Crop Yield and Acreage to Prices and Climate; KC et al., 2020: How climatic and sociotechnical factors influence crop production: a case study of canola production; Liliane and Charles, 2020: Factors Affecting Yield of Crops).
The sentence became: "This value is reasonable as the weather is among several factors (e.g. prices, sociotechnical factors, managerial decisions) affecting coffee yield (Miao et al., 2016; KC et al., 2020; Liliane and Charles, 2020)."

- L323-324: As pointed out earlier, even with smaller samples, the model can capture complex and robust weather-yield relationships. The model quality depends on many other factors such as the quality of the input data, the choice of potential predictors, the accuracy of the defined growing season etc. Please delete this sentence or support it with literature.
We removed these sentences.

- L346: Please provide an explanation of why the validation and test errors show so much variability.

We added the explanation: "These fluctuations imply that the model is overfitted, and thus, random error or noise appear. "

- L359 and L84: The reason for the selection of the case study regions should be made explicit (the selection of Cu M'gar as one district in 4 major coffee producing regions is based on a paper that is not yet published and the selection of the 10 maize producing regions in France is not explained at all).

For coffee, we removed the reference and we added the production statistic to explain our selection of the Cu M'gar district: "We focus on Cu M'gar district as it is a leading coffee-producing district in Vietnam, accounting for about 10 % of Vietnam's total coffee production (i.e. 76400 tons for the 2000-2018 average)."

For the ten maize producing regions, we mentioned in Sect. 2.1.2:

"Some specific tests (in Sect. 5) will focus on ten departments (as presented in Fig. 1(d)) where the average grain maize production is higher than $4 \times 10^5$ tons (or the area is higher than 40 thousand hectares)."

- L425: The remaining variability could also stem from other factors (see comment to L322) and change this sentence accordingly.

We changed the sentence to: "The remaining variability is rather large, and may be explained by non-climatic factors (e.g. prices, sociotechnical factors, managerial decisions, or political and social context)."

- L427: A possibility is also that the input variables do not sufficiently cover crop sensitive climatic drivers as only mean temperature and precipitation sum are considered in this study.

We changed the sentence to: "It could also come from climate; however, the model would require more detailed variables (e.g. at a daily scale) or more samples to go into deeper details of the climate-crop yield relationship."

- L434: The sentence does not make sense. Please rephrase.

We rephrased these sentences to:  "LTO indicated that a simple linear model is preferable because it has a lower testing error. "

- L444: What do you mean with "other crops will be investigated"? Afterwards you cite papers that already studied these crops I suppose. L446-447: The sentences are not easy to understand in terms of language. Please rephrase.

We rephrased these sentences to: "In addition, by using a similar approach presented here, other crops will be investigated, for instance, over France (Ceglar et al., 2016; Schauberger et al., 2018; Ceglar et al., 2020), over Europe (Ceglar et al., 2017; Lecerf et al., 2019) or globally (Bunn et al., 2015). Furthermore, these types of statistical crop models can be used to refine the potential adaptation and mitigation strategies."